# LncRNA DILA1 inhibits Cyclin D1 degradation and contributes to tamoxifen resistance in breast cancer

Qianfeng Shi[1,2,4], Yudong Li[1,2,4], Shunying Li[1,2,4], Liang Jin [1,2], Hongna Lai[1,2], Yanqing Wu[3], Zijie Cai[1,2], Mengdi Zhu[1,2], Qian Li[1,2], Ying Li[1,2], Jingru Wang[1,2], Yujie Liu[1,2], Zongqi Wu[1,2], Erwei Song [1,2] & Qiang Liu [1,2✉]

Cyclin D1 is one of the most important oncoproteins that drives cancer cell proliferation and associates with tamoxifen resistance in breast cancer. Here, we identify a lncRNA, DILA1, which interacts with Cyclin D1 and is overexpressed in tamoxifen-resistant breast cancer cells. Mechanistically, DILA1 inhibits the phosphorylation of Cyclin D1 at Thr286 by directly interacting with Thr286 and blocking its subsequent degradation, leading to overexpressed Cyclin D1 protein in breast cancer. Knocking down DILA1 decreases Cyclin D1 protein expression, inhibits cancer cell growth and restores tamoxifen sensitivity both in vitro and in vivo. High expression of DILA1 is associated with overexpressed Cyclin D1 protein and poor prognosis in breast cancer patients who received tamoxifen treatment. This study shows the previously unappreciated importance of post-translational dysregulation of Cyclin D1 contributing to tamoxifen resistance in breast cancer. Moreover, it reveals the novel mechanism of DILA1 in regulating Cyclin D1 protein stability and suggests DILA1 is a specific therapeutic target to downregulate Cyclin D1 protein and reverse tamoxifen resistance in treating breast cancer.

[1] Guangdong Provincial Key Laboratory of Malignant Tumor Epigenetics and Gene Regulation, Sun Yat-Sen Memorial Hospital, Sun Yat-Sen University, 510120 Guangzhou, China. [2] Breast Tumor Center, Sun Yat-Sen Memorial Hospital, Sun Yat-Sen University, 510120 Guangzhou, China. [3] Department of Thyroid and Breast Surgery, The First Affiliated Hospital of Sun Yat-Sen University, Sun Yat-Sen University, 510080 Guangzhou, China. [4] These authors contributed equally: Qianfeng Shi, Yudong Li, Shunying Li. ✉email: liuq77@mail.sysu.edu.cn

Sustaining proliferative signaling is one of the hallmarks of cancer[1]. CCND1 gene, which encodes the Cyclin D1 protein to drive cell proliferation, is the second most frequently amplified oncogene in human cancer across 26 histological types[2]. The interaction between Cyclin D1 and cyclin-dependent kinase 4/6 (CDK4/6), which leads to the phosphorylation and inactivation of retinoblastoma protein (Rb), is an essential regulatory step for G1–S transition and cell cycle progression[3–7].

Up to 20% of breast cancers have amplification of CCND1 gene[8–10]. Moreover, many dysregulated signaling pathways can lead to overexpressed Cyclin D1 protein in nearly 50% of breast cancers, most being estrogen receptor (ER)-positive luminal subtype. Endocrine therapy including tamoxifen is highly effective in reducing the recurrence of ER-positive early breast cancer by ~40%, but de novo or acquired resistance still occurs in one-third of such patients, leading to tumor relapse and metastasis[11]. Several clinical studies demonstrated that amplification of CCND1 gene[12] or overexpression of Cyclin D1 protein[13,14] was associated with poor prognosis in ER-positive breast cancer patients received tamoxifen. Furthermore, Cyclin D1 is still necessary for the proliferation of tamoxifen-resistant breast cancer cells because small interfering RNAs (siRNAs) targeting Cyclin D1 blocked their cell growth[15]. Alternative proliferative signaling including phosphoinositide-3 kinase (PI3K) and fibroblast growth factor receptor pathway may be responsible for upregulated Cyclin D1 after tamoxifen resistance[16]. Therefore, alternative strategy is needed to block Cyclin D1 activity in cancer. It has not been reported yet whether Cyclin D1, a tightly regulated protein with a short half-life, is dysregulated at the posttranscriptional level after tamoxifen resistance.

Long non-coding RNA (lncRNA) is a large class of transcripts from non-protein coding regions of the genome that have >200 nucleotides in length. Recent studies showed that lncRNAs regulate many important pathological processes in cancer, such as tumorigenesis, proliferation, metastasis, and drug resistance[17,18]. LncRNAs function mainly through interacting with key regulatory proteins to regulate their function or expression[19]. Nevertheless, it remains unclear whether lncRNAs directly interact with Cyclin D1 to regulate its expression or function. In this study, we investigated lncRNAs that interact with Cyclin D1 and examined their functional significance in tamoxifen resistance of breast cancer.

## Results

### Upregulated Cyclin D1 protein is responsible for tamoxifen resistance in ER-positive breast cancer cells.

Tamoxifen-resistant MCF-7 and T47D breast cancer cell lines were established as reported before (Fig. S1a, b and Zhu et al.[20]). To examine the status of Cyclin D1 expression in these tamoxifen-resistant models, quantitative PCR (qPCR) and western blotting was done to compare the mRNA and protein levels of Cyclin D1 between parental (MCF7-Pa and T47D-Pa) and resistant (MCF7-Re and T47D-Re) breast cancer cells. Western blotting showed that the protein levels of Cyclin D1 in resistant cells were significantly higher than in parental cells (Fig. 1a). However, there was no difference in Cyclin D1 mRNA levels between parental and resistant cells (Fig. S1c), indicating post-transcriptional or posttranslational changes may be responsible for the upregulated Cyclin D1 protein in tamoxifen-resistant breast cancer cells.

To determine the functional significance of upregulated Cyclin D1 protein in tamoxifen resistance, Cyclin D1 was knocked down by siRNAs in tamoxifen-resistant MCF-7 and T47D cells (Fig. S1d–f). It was found that siRNAs targeting Cyclin D1 not only restored tamoxifen sensitivity in MCF7-Re and T47D-Re cells (Fig. 1b and S1g), but also resulted in cell cycle arrest at G1

phase (Fig. S1h, i), indicating that these tamoxifen-resistant breast cancer cells are still dependent on Cyclin D1 for cell cycle progression and upregulated Cyclin D1 is responsible for their tamoxifen resistance.

### Identification of Cyclin D1-interacting long noncoding RNA 1 (DILA1).

Recently, we and other investigators have shown that lncRNAs can bind to key signaling proteins and directly regulate their signaling pathways[19,21,22]. To determine whether lncRNAs bind to Cyclin D1 and regulate its function, MCF-7 cells with exogenous HA-tagged or untagged Cyclin D1 were established and subjected to RNA immunoprecipitation (RIP) using anti-HA antibody. RIP–sequencing (RIP-seq) was then performed to identify the lncRNAs that specifically binds to HA-tagged Cyclin D1 but not to untagged Cyclin D1 control. Hierarchical clustering analysis indicated that 51 lncRNAs were significantly enriched in the RNAs pulled down from cells with HA-tagged Cyclin D1 than the cells with untagged Cyclin D1 (greater than twofold and $p < 0.05$) (Fig. 1c and Table S1). To identify lncRNAs that contribute to tamoxifen resistance, quantitative PCR with reverse transcription (RT-qPCR) showed that 6 of the 51 lncRNAs were increased both in tamoxifen-resistant MCF-7 (Fig. S2a) and T47D (Fig. S2b) cells than in parental cells. To confirm whether these lncRNAs increased in tamoxifen-resistant cells indeed bind to Cyclin D1, RIP was performed using anti-HA antibody or IgG control in MCF7-Pa cells with ectopically expressed HA-tagged Cyclin D1 and then subjected to RT-qPCR. LncD1-1, LncD1-5, and LncD1-8 were verified to specifically bind to Cyclin D1 (Fig. S2c). These three lncRNAs were then knocked down in MCF7-Re cells by respective siRNAs (Fig. S2d–f) and the sensitivity to tamoxifen was measured. It was found that only knockdown of lncD1-8 reversed tamoxifen resistance in MCF7-Re cells (Fig. S2g–i). Thus this Cyclin D1-interacting lncD1-8 that is upregulated and responsible for tamoxifen resistance is named as Cyclin D1-Interacting Long noncoding RNA 1 (DILA1).

Furthermore, RT-qPCR confirmed that DILA1 was expressed in MCF7-Pa and T47D-Pa cells and significantly upregulated in tamoxifen-resistant MCF7-Re and T47D-Re cells (Figs. 1d and S3a). Northern blotting also detected higher expression of DILA1 in MCF7-Re cells than in MCF7-Pa cells (Fig. S3b). To determine the abundance of DILA1 in breast epithelial cells, we examined the copy number of DILA1 in several breast cell lines using RT-qPCR and in vitro transcribed DILA-1 as standard. The results demonstrated that the copy numbers of DILA1 are higher in ER-positive breast cancer cells than in ER-negative breast cancer cells and immortalized breast epithelial cells (Fig. S3c, d). Moreover, the copy number of DILA1 was significantly increased in tamoxifen-resistant MCF-7 and T47D cells ($385 \pm 22$ and $254 \pm 22$ copies per cell, respectively) than in the parental ones ($130 \pm 28$ and $75 \pm 5$ copies per cell, respectively; Fig. S3d). Importantly, RIP-qPCR demonstrated that immunoprecipitation with anti-Cyclin D1 antibody specifically retrieved DILA1 in MCF7-Re cells, indicating that DILA1 binds to Cyclin D1 protein in an endogenous setting (Fig. 1f).

In UCSC (University of California Santa Cruz) Genome Browser, DILA1 is labeled as a long intergenic noncoding RNA named as ENST00000435697.1, which is an intergenic transcript of MIR99AHG. To determine the exact sequence of DILA1, we performed 5′ and 3′ RACE (rapid amplification of complementary DNA ends) based on the 716 nucleotides (nt) sequence in UCSC Genome Browser. The RACE results indicate that the full length of DILA1 is 1183 nt (Fig. 1f and Table S2). Furthermore, the lack of protein-coding potential of DILA1 was confirmed by ORF (open reading frame) Finder, validating DILA1 as an

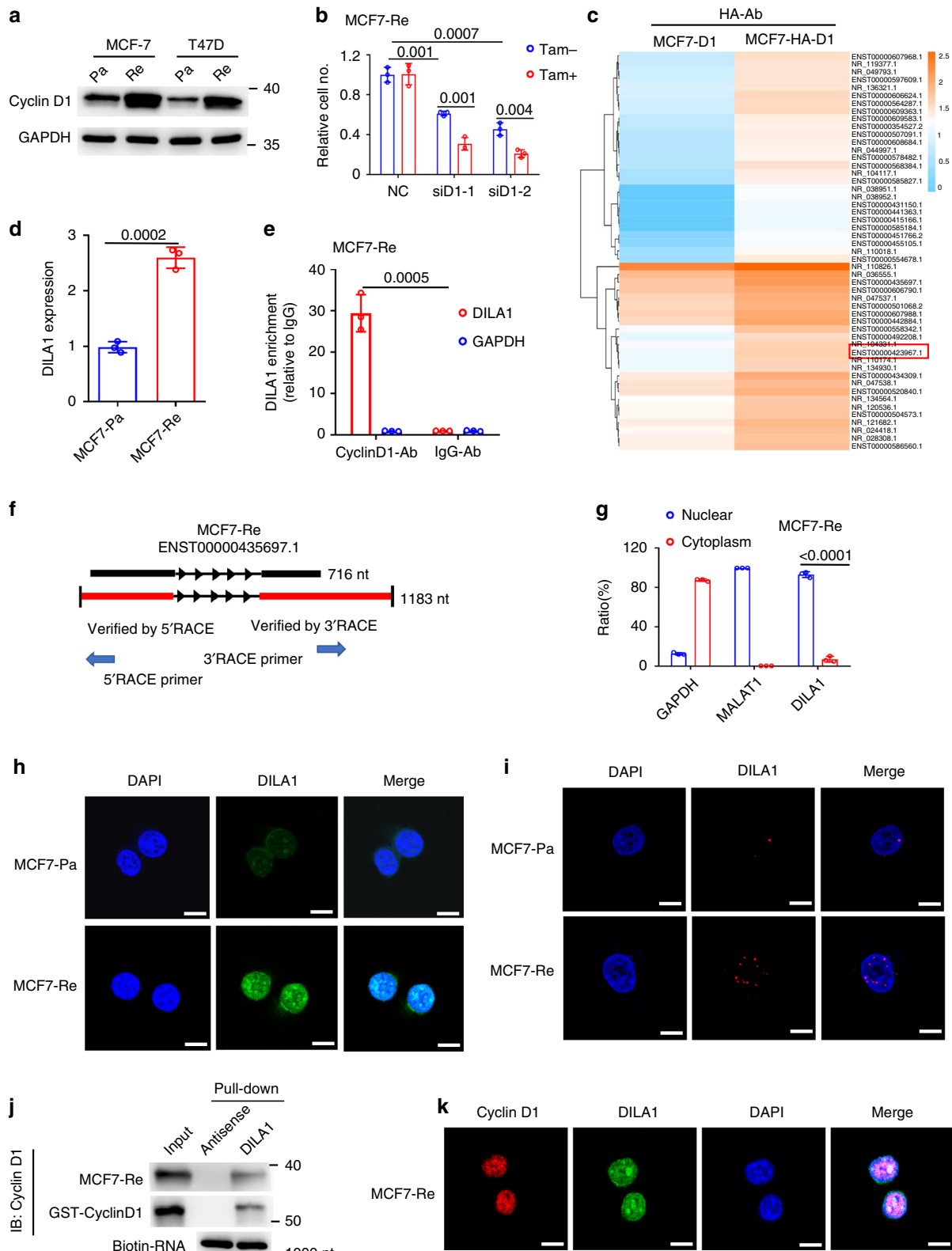

lncRNA. To determine the subcellular localization of DILA1, nuclear and cytoplasmic fractionation of MCF7-Re cells was performed and showed that the majority of DILA1 localized in the nucleus, similar to MALAT1 (Fig. 1g). Confocal microscopy of fluorescence in situ hybridization (FISH) (Fig. 1h) and RNAScope assay (Figs. 1i and S3e) also demonstrated that DILA1 located primarily in the nucleus and was upregulated in tamoxifen-resistant cells, suggesting that DILA1 may exert its biological function mainly in the nucleus.

To explore whether DILA1 directly interacts with Cyclin D1, we performed RNA pull-down assays with biotin-labeled DILA1 in MCF7-Re cells lysates and found that only DILA1, but not its antisense, pulled down Cyclin D1 protein, which was further confirmed with in vitro RNA pull-down of purified recombinant

**Fig. 1 A Cyclin D1-interacting lncRNA DILA1 is overexpressed in tamoxifen-resistant breast cancer cells with upregulated Cyclin D1 protein. a** Western blotting showing Cyclin D1 protein in parental and tamoxifen-resistant MCF-7 (MCF7-Pa, MCF-Re) and T47D (T47D-Pa, T47D-Re) cells. GAPDH was used as a loading control. **b** MCF-Re cells, transfected with negative control siRNA (NC) or one of the two siRNAs targeting Cyclin D1 (siD1-1 and siD1-2), were treated with tamoxifen for 48 h. Relative cell numbers were determined by a cell counter. **c** Heatmap of the enriched long noncoding RNAs by RIP–sequencing in MCF-7 cells with ectopically expressed HA-Cyclin D1 and untagged Cyclin D1 control. (fold change >2, $p < 0.05$). $p$ values were determined by negative binomial generalized linear models. No adjustments were made for multiple comparisons. **d** RT-qPCR showing the expression of DILA1 in MCF7-Pa and MCF-Re cells. **e** Binding of DILA1 to CyclinD1 protein in MCF7-Re cells, assayed by RIP, followed by RT-qPCR. IgG and GAPDH were used as negative controls. **f** The full length of DILA1 (ENST00000435697.1) in UCSC Genome Browser (upper) and determined by 5′ and 3′ RACE (lower). **g** RT-qPCR showing the nuclear and cytoplasmic fraction of DILA1 in MCF-Re cells, with GAPDH and MALAT1 as cytoplasmic and nuclear control, respectively. **h** Confocal FISH images showing nuclear localization of DILA1 (green) in MCF7-Pa and MCF-Re cells. **i** RNAScope showing subcellular localization and relative expression of DILA1 (red) in MCF7-Pa and MCF7-Re cells. **j** RNA pull-down showing the interaction between Cyclin D1 and DILA in vitro (MCF7-Re cell lysates or recombinant GST-Cyclin D1 protein). Biotin-labeled DILA1 detection by anti-biotin antibody as a control. **k** Confocal FISH images showing the co-localization of Cyclin D1 (red) and DILA1 (green) in MCF7-Re cells. For **a**, **h**–**k**, representative images of three biologically independent experiments are shown. For **b**, **d**, **e**, **g**, $n = 3$ biologically independent experiments, means ± s.d. are shown, and $p$ values were determined by two-tailed Student's test. For **h**, **i**, **k**, scale bars represented 10 μm.

Cyclin D1 in a cell-free system (Fig. 1j), indicating the direct interaction between DILA1 and Cyclin D1 protein. Confocal microscopy also showed the co-localization of DILA1 with Cyclin D1 in the nucleus of MCF7-Re cells (Fig. 1k).

**DILA1 promotes cell proliferation and tamoxifen resistance.** Since DILA1 mainly localizes in the nucleus and siRNAs were reported to be less efficient than antisense oligonucleotides (ASOs) for modulating nuclear-localized lncRNAs[23], DILA1 was knocked down by ASOs in tamoxifen-resistant MCF7 and T47D cells to examine its function. RT-qPCR showed that both ASOs efficiently reduced DILA1 expression in MCF7-Re and T47D-Re cells (Fig. S4a, b). Knocking down of DILA1 by ASOs not only slowed down the cell growth but also restored their sensitivity to tamoxifen in MCF7-Re and T47D-Re cells, demonstrated by 3-[4,5-dimethylthiazol-2-yl]-2,5 diphenyl tetrazolium bromide (MTT) assay (Figs. 2a and S4c), colony-formation assay (Figs. 2b and S4d) and 5-ethynyl-2′-deoxyuridine (EdU) incorporation by fluorescence microscopy (Fig. 2c). Moreover, cell cycle analysis by flow cytometry showed that DILA1-ASOs caused G1/S cell cycle arrest in MCF7-Re and T47D-Re cells, which was further increased by tamoxifen (Figs. 2d and S4e). These results suggest that DILA1 is necessary for cell proliferation and tamoxifen resistance.

To determine whether DILA1 is sufficient to drive cell proliferation and cause tamoxifen resistance, DILA1 was ectopically expressed in parental MCF7 and T47D cells by transfecting with PCDH-puro expression vector carrying the DILA1 sequence (Fig. S4f, g). Consistent with the results of DILA1-ASOs, overexpression of DILA1 in MCF7-Pa and T47D-Pa cells promoted cell proliferation and tamoxifen resistance (Figs. 2e–g and S4h, i). DILA1 accelerated cell cycle progression by decreasing the percentage of G1 cells, which was not affected by tamoxifen (Figs. 2h and S4j). Together, these results indicate that DILA1 is not only necessary but also sufficient to promote cell proliferation and cause tamoxifen resistance.

**Decreased degradation is responsible for upregulated Cyclin D1 protein in tamoxifen-resistant cells.** To study at which level Cyclin D1 protein was dysregulated in tamoxifen-resistant cells, western blotting was done to measure the protein expression of Cyclin D1 in parental and tamoxifen-resistant MCF-7 and T47D cells before and after tamoxifen treatment at different time points. It was found that Cyclin D1 protein remained steady without tamoxifen treatment and significantly decreased after tamoxifen treatment in MCF7-Pa and T47D-Pa cells (Figs. 3a and S5a), whereas Cyclin D1 protein in MCF7-Re and T47D-Re cells accumulated over time, regardless of tamoxifen treatment

(Figs. 3b and S5b). However, there was no significant change at the levels of Cyclin D1 mRNA in resistant cells over time (Fig. S5c, d). The upregulated Cyclin D1 protein with similar mRNA level in tamoxifen-resistant cells compared to parental cells indicate that either translational or posttranslational dysregulation may be responsible for the increased Cyclin D1 protein in tamoxifen-resistant cells.

To further elucidate the mechanism of Cyclin D1 upregulation, the levels of Cyclin D1 protein in parental and resistant breast cancer cells were examined when cycloheximide (CHX) or MG132 was used to inhibit de novo protein synthesis or proteasome degradation respectively. When protein synthesis was inhibited by CHX, Cyclin D1 protein decreased rapidly and its half-life was ~1 h in MCF7-Pa and T47D-Pa cells, but Cyclin D1 protein remained high and its half-life significantly increased to >2 h in MCF7-Re and T47D-Re cells. (Figs. 3c and S5e), indicating that the protein stability of Cyclin D1 in tamoxifen-resistant cells is increased. However, when proteasome degradation was inhibited by MG132 for ~2 h to see whether there is a difference of protein synthesis, Cyclin D1 levels remained steady in MCF7-Re and MCF7-Pa cells (Fig. 3d), indicating that the protein synthesis of Cyclin D1 was not different between MCF7-Re and MCF7-Pa cells. Together with similar Cyclin D1 mRNA levels in MCF7-Re and MCF7-Pa cells, these results suggest that protein degradation but not protein synthesis was responsible for the upregulated Cyclin D1 protein in tamoxifen-resistant breast cancer cells.

**DILA1 inhibits Cyclin D1 degradation via ubiquitin–proteasome pathway.** To study the mechanism that Cyclin D1-interacting DILA1 leads to tamoxifen resistance, the expression of DILA1 was knocked down with ASOs in MCF7-Re cells in the absence or presence of CHX or MG132. It was found that Cyclin D1 protein was significantly downregulated by DILA1 ASOs but remained high in the presence of MG132 (Fig. 3e). Nevertheless, DILA1-ASOs markedly decreased Cyclin D1 protein expression in the presence of CHX treatment, with the half-life of Cyclin D1 decreased from >2 h to <1 h in MCF7-Re cells (Fig. 3f), suggesting that DILA1-ASOs accelerated the degradation of Cyclin D1 protein in MCF7-Re cells that overexpressed DILA1. Consistent with the results of DILA1-ASOs in MCF7-Re cells, overexpression of DILA1 in MCF7-Pa cells significantly increased the Cyclin D1 protein expression and extended the half-life of Cyclin D1 protein from <1 h to ~2 h under CHX treatment (Fig. 3g). These results indicate that DILA1 increases the protein expression of Cyclin D1 by inhibiting its degradation.

The ubiquitin–proteasome pathway is responsible for the degradation and recycling of many proteins[24–26], including Cyclin D1[27–29]. Immunoprecipitation of Cyclin D1 followed by

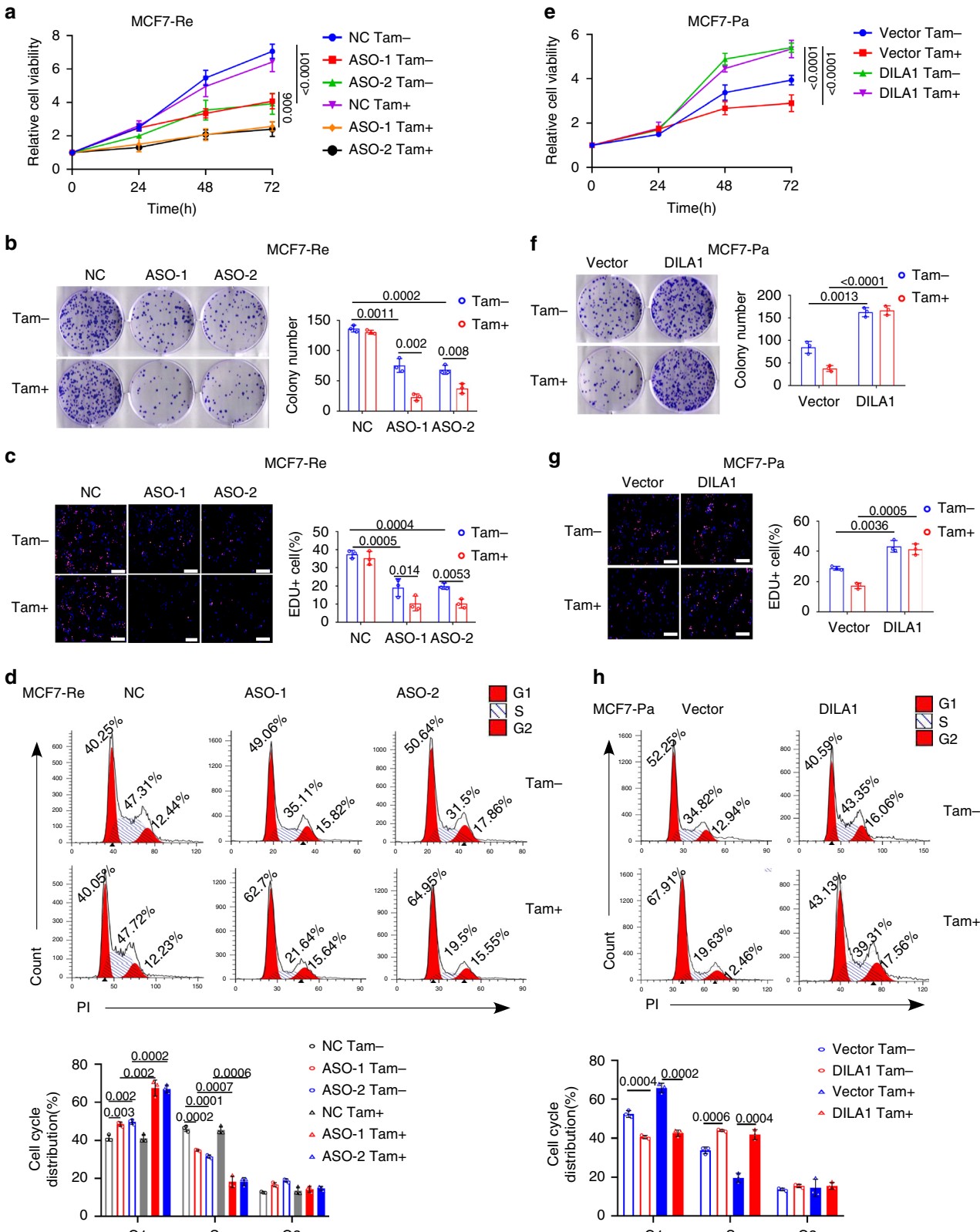

**Fig. 2 DILA1 promotes cell proliferation and tamoxifen resistance. a–d** MCF-Re cells were transfected with NC or ASOs targeting DILA1 and then treated with 3 µM tamoxifen (Tam). **a** MTT assay showing relative cell growth at 0, 24, 48, and 72 h. Representative images of colony formation (**b**) or EdU incorporation by fluorescence microscopy (**c**). Flow cytometry showing the cell cycle distribution of cells (**d**). **e–h** MCF7-Pa cells were transfected with control vector or vector expressing DILA1 and then treated with 3 µM tamoxifen (Tam). MTT assay showing relative cell growth at 0, 24, 48, and 72 h (**e**). Representative images of colony formation (**f**) or EdU incorporation by fluorescence microscopy (**g**). Flow cytometry showing the cell cycle distribution of cells (**h**). For **c**, **g**, scale bars represented 100 µm. For all experiments, n = 3 biologically independent experiments. For **a–h**, means ± s.d. were shown, and p values were determined by two-tailed Student's test.

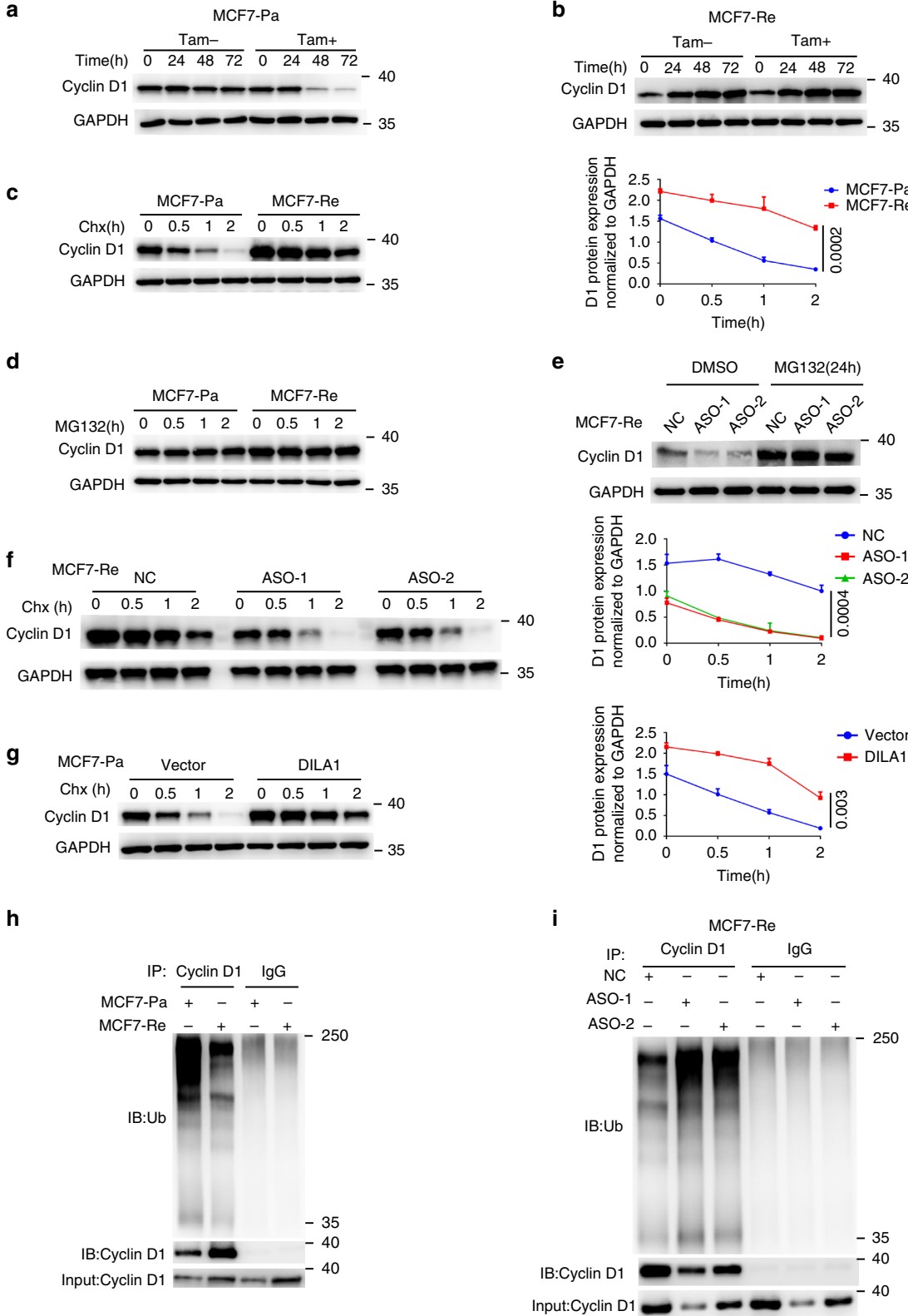

anti-ubiquitin immunoblotting demonstrated that ubiquitinated Cyclin D1 was markedly decreased in MCF7-Re and T47D-Re cells than in MCF7-Pa and T47D-Pa cells (Figs. 3h and S5g). Moreover, DILA1-ASOs significantly increased ubiquitinated Cyclin D1 in MCF7-Re cells (Fig. 3i), suggesting that DILA1 inhibits the ubiquitination of Cyclin D1 and leads to upregulated Cyclin D1 protein in tamoxifen-resistant breast cancer cells.

**DILA1 inhibits the phosphorylation (Thr286) of Cyclin D1 by directly interacting with Thr286 and blocks nuclear-to-cytoplasmic redistribution of Cyclin D1 in tamoxifen-resistant cells.** It has been reported that the phosphorylation of threonine-286 (Thr286) at Cyclin D1 by glycogen synthase kinase 3β (GSK3β) is required for the ubiquitination and degradation of Cyclin D1[27,30,31]. Indeed, phosphorylated Cyclin D1 at Thr286,

**Fig. 3 DILA1 inhibits Cyclin D1 degradation via the ubiquitin–proteasome pathway. a, b** Western blotting showing Cyclin D1 protein in MCF7-Pa (**a**) and MCF7-Re (**b**) cells treated with tamoxifen. **c** Western blotting showing Cyclin D1 protein in MCF7-Pa and MCF7-Re cells treated with CHX for the indicated time (left). The quantification of Cyclin D1 degradation rate by gray scale analysis (right). **d** Western blotting showing Cyclin D1 protein in MCF7-Pa and MCF7-Re cells treated with MG132 for the indicated time. **e** Western blotting showing Cyclin D1 protein in MCF7-Re cells transfected with NC or DILA-ASOs and then treated with MG132 for 24 h. **f** Western blotting showing Cyclin D1 protein in MCF7-Re cells transfected with NC or DILA-ASOs and then treated with CHX for the indicated time (left). The quantification of Cyclin D1 degradation rate by gray scale analysis (right). **g** Western blotting showing Cyclin D1 protein in MCF7-Pa cells transfected with control vector or vector expressing DILA1 and then treated with CHX (left). The quantification of Cyclin D1 degradation rate by gray scale analysis (right). **h** Ubiquitinated Cyclin D1 detected by immunoprecipitation with anti-Cyclin D1 antibody or IgG control and immunoblotting with anti-ubiquitin antibody in MCF7-Pa and MCF-Re cells. **i** Ubiquitinated Cyclin D1 detected by immunoprecipitation with anti-Cyclin D1 antibody or IgG control and immunoblotting with anti-ubiquitin antibody in MCF-Re cells transfected with NC or DILA-ASOs. For **a–i**, GAPDH was used as a loading control. Representative images of three biologically independent experiments are shown. For **c**, **f**, **g**, means ± s.d. are shown, and p values were determined by two-tailed Student's test.

but not the one at Ser90, was significantly decreased in MCF7-Re and T47D-Re cells than in MCF7-Pa and T47D-Pa cells (Figs. 4a and S6a). Additionally, there was no difference in the expression of GSK3β between parental and resistant cells (Figs. 4a and S6a). Furthermore, DILA1-ASOs markedly decreased total Cyclin D1 and increased Thr286-phosphorylated Cyclin D1 in MCF7-Re cells (Fig. 4b), while overexpression of DILA1 increased total Cyclin D1 and decreased Thr286-phosphorylated Cyclin D1 in MCF7-Pa cells (Fig. 4c). It was shown that the expression of GSK3β was not changed by DILA1 both in MCF7-Pa and MCF7-Re cells (Fig. 4b, c). Furthermore, when constitutively activated GSK3β (GSK3β-CA) was transfected into MCF7-Pa cells, phosphorylated Cyclin D1 (Thr286) increased, which was reversed by exogenous DILA1, indicating that DILA1 regulates Cyclin D1 phosphorylation at the level downstream of GSK3β (Fig. 4d).

It was reported that GSK3β-dependent Thr286 phosphorylation led to the nuclear-to-cytoplasmic redistribution of Cyclin D1 to interact with E3 ligase in cytoplasm for degradation[27,30,31]. Indeed, nuclear and cytoplasm fractionation showed more nuclear Cyclin D1 in MCF7-Re cells than in MCF7-Pa cells, supporting less degradation of Cyclin D1 in tamoxifen-resistant cells (Fig. 4e). Furthermore, DILA1-ASOs in MCF7-Re cells significantly decreased Cyclin D1 in the nucleus and increased Cyclin D1 in the cytoplasm (Fig. 4f). Consistently, overexpressed DILA1 in MCF7-Pa cells significantly decreased Cyclin D1 in cytoplasm and increased Cyclin D1 in nucleus (Fig. 4g). Collectively, these data suggest that DILA1 inhibits the phosphorylation (Thr286) of Cyclin D1 and the ensuing nuclear-to-cytoplasm translocation.

Cyclin D1 and CDK4/6 play a crucial role in G1/S cell cycle progression by phosphorylating and inactivating the Rb, a tumor suppressor that restrains G1/S cell cycle transition[32]. We found a higher level of Ser780-phosphorylated Rb (p-Rb(Ser780)) in tamoxifen-resistant MCF-7 (Fig. S6b) and T47D (Fig. S6c) cells than in the parental ones, while total Rb protein levels were similar. Moreover, DILA1-ASOs in MCF7-Re cells or over-expressed DILA1 in MCF7-Pa cells decreased or increased the p-Rb(Ser780) levels, respectively, without changing the total Rb levels (Fig. S6d, e). Together, these results indicate that DILA1 enhances the classical Ser780 phosphorylation of Rb and accelerates G1/S cell cycle progression.

To identify the exact sequence of DILA1 that binds Cyclin D1, a series of DILA1 deletion mutants were constructed (Fig. S6f) and examined for their binding to Cyclin D1 by RNA pull-down assay with MCF7-Re cell lysates or purified recombinant Cyclin D1.

It was found that the DILA1 mutants missing the 1000–1183 nt lost the binding to Cyclin D1 and the 1000–1183 nt of DILA1 showed the affinity to bind Cyclin D1 comparable to full-length DILA1, suggesting that the sequence of 1000–1183 nt

in DILA1 is essential and sufficient for the interaction between DILA1 and Cyclin D1 (Fig. 4h). Moreover, enhanced crosslinking IP and qPCR (eCLIP-qPCR) was performed to identify the lncRNA fragment that bound and was protected by Cyclin D1 from RNAase digestion. Among the 11 pairs of primers, only the primer pair 11 designed for the 1000–1183 nt region of DILA1 successfully amplified DILA1 segment in eCLIP-qPCR, confirming that the 1000–1183 nt of DILA1 was the main region responsible for binding with Cyclin D1 (Fig. 4i). To determine the functional role of the 1000–1183 nt of DILA1, the last part of DILA1 (1000–1183 nt, DILA1-S6) was ectopically expressed in MCF7-Pa cells. It was found that DILA1-S6 inhibited the phosphorylation of Cyclin D1 at Thr286, decreased the nuclear-to-cytoplasm translocation and ubiquitination of Cyclin D1, and caused tamoxifen resistance (Fig. S7).

The RNAfold software predicts that a stable hairpin (1000–1046 nt, hairpin A) structure exists within 1000–1183 nt of DILA1 (Fig. S6g). Indeed, RNA pull-down assay with MCF7-Re cell lysates or purified recombinant Cyclin D1 showed that mutation of hairpin A in DILA1 (DILA1-mA) completely abolished the binding between DILA1 and Cyclin D1 (Fig. 4j), suggesting that hairpin A was the site of DILA1 that directly interacts with Cyclin D1.

There are three functional domains in Cyclin D1 protein, the p-Rb interaction domain, the Cyclin box domain, and the PEST domain[33]. To determine the exact domain of Cyclin D1 bound to DILA1, vectors carrying HA-tagged full length or truncation mutants of Cyclin D1 (D1-FL (1–295), D1-T1 (20–295), D1-T2 (91–295), and D1-T3 (1–256)) were constructed (Fig. S6h). RNA pull-down assay was performed using in vitro synthesized biotin-labeled full-length DILA1 to examine its interaction with different constructs of Cyclin D1 ectopically expressed in MCF7-Pa cells. D1-FL, D1-T1, and D1-T2, but not D1-T3 truncation mutants that lack the "PEST" sequence, were pulled down by DILA1 (Fig. 4k), suggesting that DILA1 mainly interacts with the 257–296 residues of Cyclin D1 that contain a "PEST" motif. This interaction could protect Cyclin D1 from degradation because the "PEST" domain is crucial to its ubiquitination and degradation[34].

To determine whether DILA1 can directly inhibit the phosphorylation of Cyclin D1 by GSK3β, purified inactive (GSK3β-KD) and constitutively active form (GSK3β-CA) of GSK3β[30] was added to purified recombinant Cyclin D1 protein in the absence or presence of DILA1 or DILA1-mA. In vitro phosphorylation assay showed that GSK3β-CA, but not GSK3β-KD, efficiently phosphorylated Cyclin D1 at Thr286. Furthermore, DILA1, but not DILA1-mA, significantly inhibited the phosphorylation of Cyclin D1 by GSK3β (Fig. 4l). More interestingly, RIP assay showed that immunoprecipitation of HA-tagged Cyclin D1, but not HA-tagged Cyclin D1 mutant at threonine 286 (T286A), enriched DILA1 (Fig. 4m), suggesting

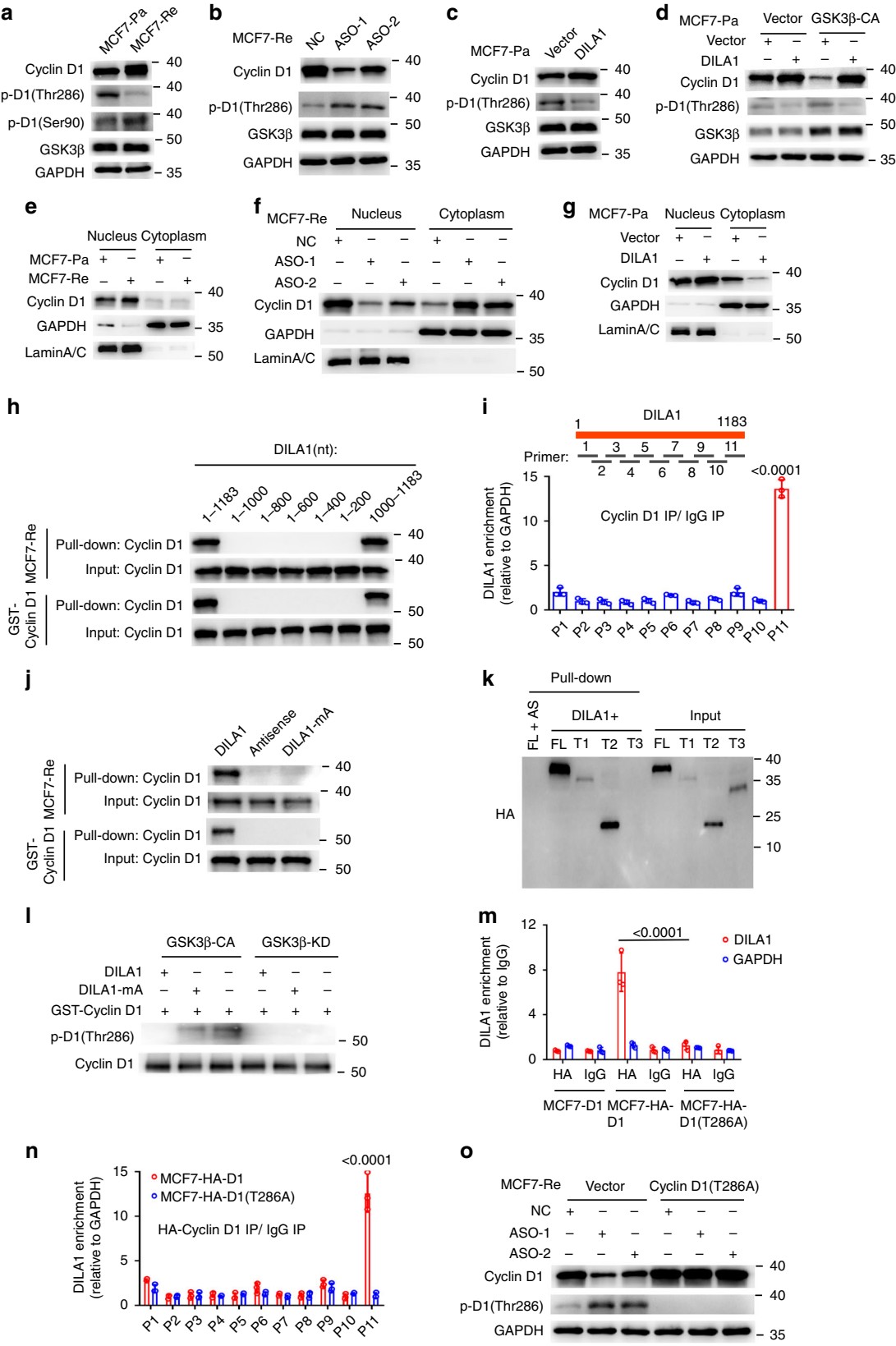

that the threonine at 286 is the site of Cyclin D1 that directly interacts with DILA1. eCLIP-qPCR assays were performed using an anti-HA antibody in MCF7-Re cells transfected with vectors expressing HA-Cyclin D1 or HA-Cyclin D1(T286A). As expected, only the primer pair 11 amplified DILA1 segment from the precipitates in cells with HA-tagged Cyclin D1 but not

in cells with HA-tagged Cyclin D1 mutant at threonine 286 (T286A) (Fig. 4n).

Furthermore, it was found that DILA1-ASOs had no effect on the expression of Cyclin D1 T286A that was ectopically expressed in MCF7-Re cells (Fig. 4o). In total, these results indicate that the hairpin A of DILA1 directly interacts with the Thr286 at Cyclin

**Fig. 4 Hairpin A of DILA1 interacts with Thr286 of Cyclin D1 and inhibits its phosphorylation and nuclear-to-cytoplasmic redistribution. a** Western blotting showing the levels of Cyclin D1, phosphorylated Cyclin D1 (p-D1) (Thr286), p-D1 (Ser-90), and GSK3β in MCF7-Pa and MCF-Re cells. **b–d** Western blotting showing the levels of Cyclin D1, p-D1(Thr286), and GSK3β in DILA1-silenced MCF-Re cells (**b**), in DILA1-overexpressed MCF7-Pa cells (**c**), and in DILA1 and constitutively activated GSK3β (GSK3β-CA) both overexpressing MCF7-Pa cells (**d**). GAPDH was used as a loading control. **e–g** Western blotting showing the levels of nuclear and cytoplasmic Cyclin D1 in MCF7-Pa and MCF-Re cells (**e**), in DILA1-silenced MCF-Re cells (**f**), and in DILA1-overexpressed MCF7-Pa cells (**g**). GAPDH and Lamin A/C was used as cytoplasmic and nuclear controls, respectively. **h** RNA pull-down showing the interaction between Cyclin D1 and full-length or serial truncation mutants of DILA1. **i** eCLIP-qPCR assay indicating the exact DILA1 region responsible for Cyclin D1 binding in MCF7-Re cells (bottom). A schematic diagram of DILA1 primers (P1–P11) designed for eCLIP-qPCR, covering the full length of DILA1 (top). **j** RNA pull-down showing the interaction between Cyclin D1 and DILA1 or DILA mutant with deletion of hairpin A (DILA1-mA). **k** RNA pull-down showing the interaction between DILA1 and HA-tagged full length or truncation mutants (FL (1–295), T1 (20–295), T2 (91–295), and T3 (1–256)) of Cyclin D1 proteins in MCF-7 cells. AS (antisense) was used as a negative control. **l** In vitro kinase assay showing the phosphorylation of Cyclin D1 at Thr286 by active GSK3β-CA or inactive GSK3β-KD in the absence or presence of DILA1 or DILA1-mA. **m** RIP-qPCR of DILA1 immunoprecipitated with anti-HA antibody or IgG in MCF7-Re cells with ectopically expressed HA-Cyclin D1 or HA-Cyclin D1(T286A). **n** eCLIP-qPCR assay in MCF7-Re cells with HA-Cyclin D1 or HA-Cyclin D1(T286A) overexpressed. For **i**, **m**, **n**, $n = 3$ biologically independent experiments, means ± s.d. are shown, and $p$ values were determined by two-tailed Student's test. **o** Western blotting showing the levels of Cyclin D1 and p-D1(Thr286) in MCF7-Re cells transfected with control vector or vector expressing Cyclin D1(T286A) and then transfected with NC or DILA1-ASOs. For **a–h**, **j–l**, **o**, representative images of three biologically independent experiments are shown.

D1, blocking its phosphorylation and subsequent ubiquitination/degradation.

**DILA1 promotes tamoxifen resistance in vivo.** To investigate the role of DILA1 in regulating tamoxifen resistance of ER-positive breast cancer in vivo, MCF7-Re cells were inoculated into the mammary fat pads of NOD/SCID mice. Mice were treated with tamoxifen or control when tumors became palpable. DILA1-ASOs or control was then injected into tumors every 2 days. Consistent with the results in vitro, DILA1-ASOs significantly decreased tumor volumes than control oligos, and tamoxifen treatment further shrunk the tumors, suggesting that DILA1-ASOs inhibits the tumor growth and restores the sensitivity to tamoxifen in tamoxifen-resistant tumors (Fig. 5a, b). Immunohistochemistry (IHC) staining showed that the expression of Ki67, Cyclin D1, and p-Rb(Ser780) were markedly lower in tumors with DILA1-ASO treatment and were further decreased by tamoxifen treatment, while the expression of Thr286-phosphorylated Cyclin D1 (p-D1(Thr286)) was increased with DILA1-ASO treatment, and the efficiency of DILA1 knockdown was confirmed by ISH (Fig. 5c–h). These results indicate that antagonizing DILA1 significantly decreases cancer cell proliferation, inhibits Cyclin D1 protein expression and its downstream Rb protein phosphorylation, and reverses tamoxifen resistance in vivo.

MCF7-Pa cells with stably overexpressed DILA1 or vector control by lentiviral infection were inoculated into the mammary fat pads of NOD/SCID mice. Tumors with overexpressed DILA1 were significantly larger than vector control and resistant to tamoxifen treatment (Fig. S8a, b). IHC staining and ISH staining showed that overexpression of DILA1 led to a markedly higher expression of Ki67, Cyclin D1, and p-Rb(Ser780) but lower p-D1 (Thr286) levels in the tumor that were not affected by tamoxifen (Fig. S8c–h).

**High DILA1 expression is associated with higher Cyclin D1 protein expression, tamoxifen resistance, and poor prognosis in ER-positive breast cancer patients.** To determine whether the findings above are clinically relevant, the expression of DILA1 and Cyclin D1, p-D1(Thr286), p-Rb(Ser780), and Ki67 protein was evaluated in the primary tumor samples from 190 ER-positive breast cancer patients who received adjuvant tamoxifen treatment. The cut-off value of staining intensity (staining index (SI)) score to determine high or low DILA1 expression was calculated by receiver operating characteristic curve and it indicated

that SI = 3 was the optimal score to separate the DILA1-high and DILA1-low groups (Fig. S9a). The DILA1-high group exhibited higher Cyclin D1, p-Rb(Ser780), and Ki67 expression but lower p-D1(Thr286) expression, whereas the DILA1-low group showed lower expression of Cyclin D1, p-Rb(Ser780), and Ki67 but higher expression of p-D1(Thr286) (Fig. 6a–e and Table S4). Collectively, these results indicate that DILA1 regulates the expression of Cyclin D1 and subsequent Rb phosphorylation in vivo.

Then the correlation between the expression of DILA1 or Cyclin D1 and the clinicopathological parameters was analyzed. Higher DILA1 expression was significantly associated with clinical stage, lymph node metastasis, and Ki67 staining ($p < 0.05$) but not age, tumor size, or Her-2 status (Table S5). Higher Cyclin D1 expression was significantly associated with the level of Ki67 expression but not with other factors analyzed in this patient cohort (Table S6). ISH and IHC staining showed that the expression of DILA1, Cyclin D1, and p-Rb(Ser780) protein was significantly higher in the samples from relapsed patients than from non-relapsed patients (Figs. 6f, g and S9b). Importantly, Spearman rank correlation analysis showed a positive correlation between DILA1 and Cyclin D1 protein in the patients' samples ($r = 0.57$, $p < 0.0001$; Fig. S9c).

Kaplan–Meier survival curve analysis (K-M analysis) indicated that higher DILA1, Cyclin D1, and p-Rb(Ser780) protein expression in these ER-positive breast cancer patients was associated with shorter relapse-free survival (Figs. 6h, i and S9d). Then we performed K-M analysis to examine the association between the overall survival and the expression of DILA1, Cyclin D1, and p-Rb. The results showed that Cyclin D1, DILA1, or p-Rb, were not significantly associated with the overall survival, although CyclinD1 ($p = 0.064$) and DILA1 ($p = 0.073$) showed a statistically insignificant trend (Fig. S9e–g). This may be caused by the relatively small patient number in this cohort, different treatments after relapse, etc. Moreover, univariate and multivariate Cox proportional hazards regression analysis showed that only DILA1 expression was an independent prognosis predictor in ER-positive breast cancer (Table S7). Collectively, increased expression of DILA1, Cyclin D1, and p-Rb(Ser780) protein are associated with tamoxifen resistance and poor prognosis of ER-positive breast cancer patients who received adjuvant tamoxifen treatment, consistent with the findings in tamoxifen-resistant breast cancer cell lines.

**Discussion**

In this study, we identified an lncRNA named DILA1 that specifically binds to the Thr286 of Cyclin D1 protein and inhibited

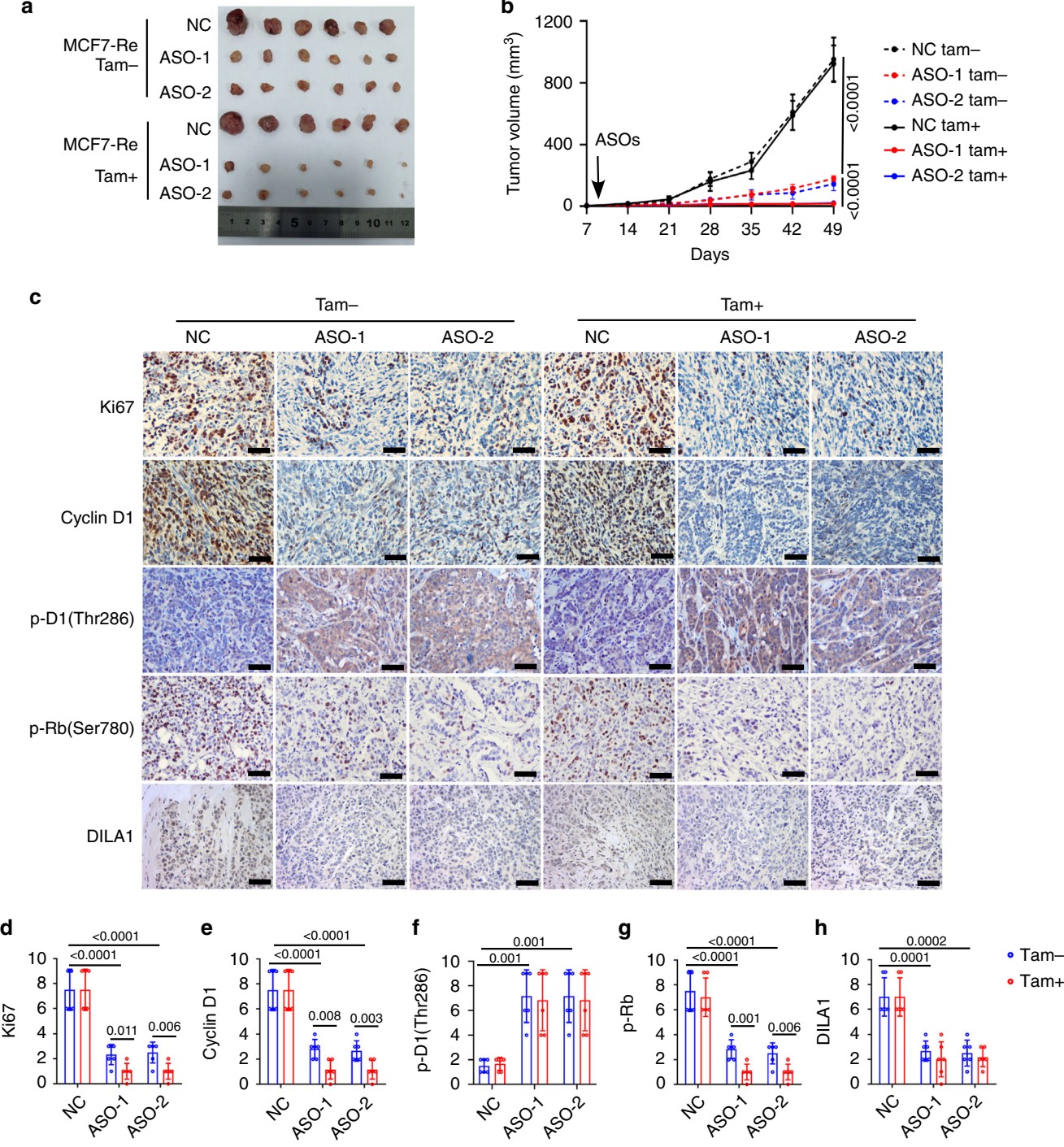

**Fig. 5 DILA1 promotes tamoxifen resistance in vivo.** MCF7-Re cells were inoculated into the mammary fat pads of NOD/SCID mice. When tumors became palpable, MCF7-Re tumors were injected with NC or DILA1-ASOs intratumorally. The tumor picture (**a**) and the tumor growth curves (**b**) are shown and compared among the groups. **c** Immunohistochemistry (IHC) staining of Ki67, Cyclin D1, p-D1(Thr286), and p-Rb(Ser780) and in situ hybridization (ISH) staining of DILA1 in the tumors. Representative images of six xenografts from each group are shown. **d**–**h** The quantification of IHC scores of Ki67(**d**), Cyclin D1(**e**), p-D1(Thr286) (**f**), and p-Rb(Ser780) (**g**) and ISH scores of DILA1 (**h**). For **b**, **d**–**h**, $n = 6$ mice per group, means ± s.d. are shown, and $p$ values were determined by two-tailed Student's test. Scale bars represents 50 μm.

its phosphorylation, leading to decreased ubiquitination and degradation of Cyclin D1 (Fig. S10). The subsequent upregulated Cyclin D1 protein accelerated cell proliferation and resulted in tamoxifen resistance in breast cancer cells. Knocking down DILA1 decreased the expression of Cyclin D1 protein and reversed tamoxifen resistance of breast cancer cells both in vitro and in vivo. More importantly, high expression of DILA1 was

associated with overexpressed Cyclin D1 protein and poor prognosis in ER+ breast cancer patients who received tamoxifen.

Excessive cell proliferation is one of the hallmarks of cancer. Cyclin D1 protein, one of the most important oncoproteins in human cancer, accelerates cell proliferation by engaging CDK4/6 to phosphorylate Rb and drive G1–S transition. It has been well established that overexpressed Cyclin D1 protein is caused by the

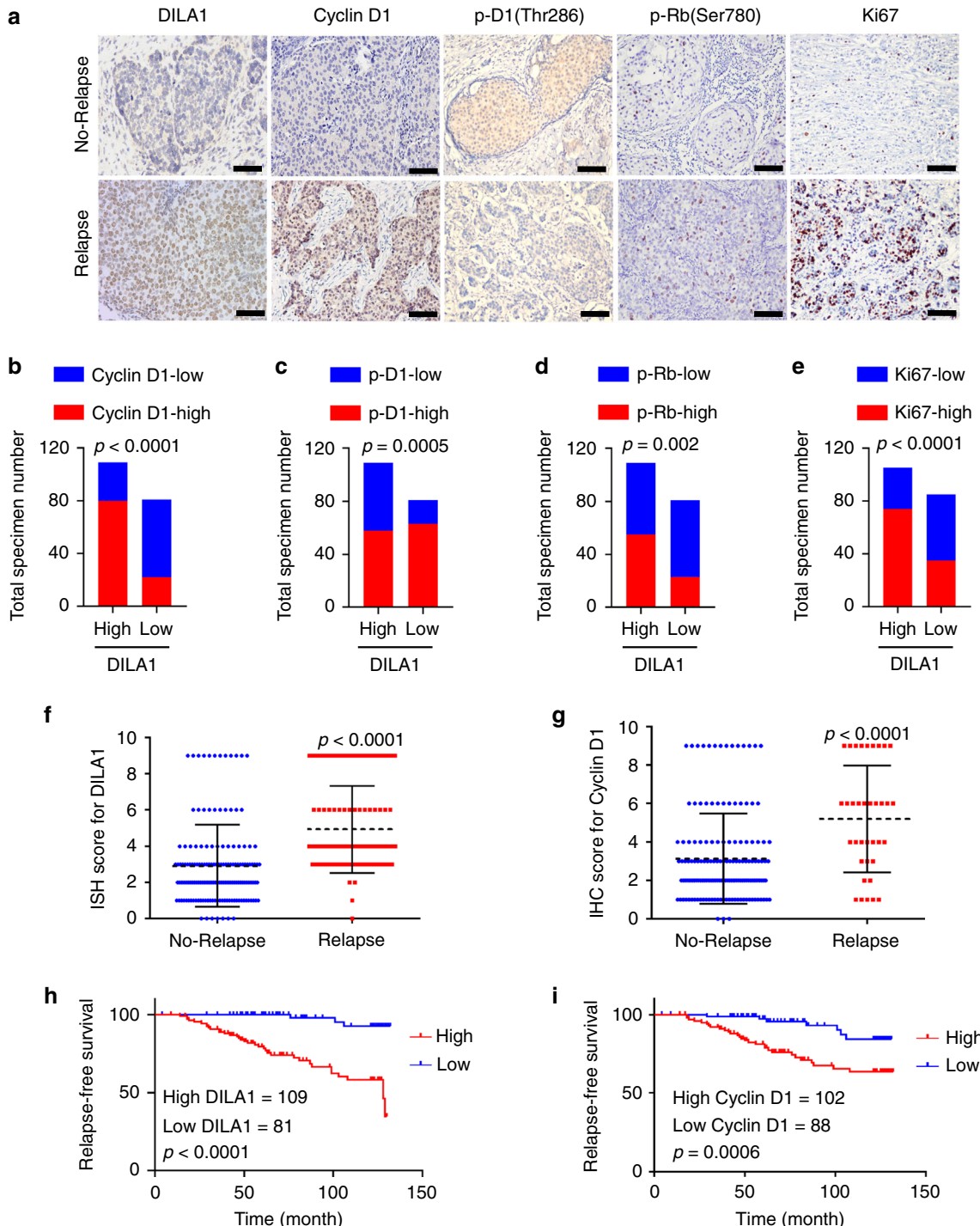

**Fig. 6 High DILA1 expression is associated with higher Cyclin D1 protein expression, tamoxifen resistance, and poor prognosis in ER-positive breast cancer patients. a** ISH staining for DILA1 and IHC staining for Cyclin D1, p-D1(Thr286), p-Rb(Ser780), and Ki67 in breast cancer specimen from ER-positive patients with or without relapse. Representative images of tumor specimens from ER-positive patients with ($n = 36$) or without relapse ($n = 154$) are shown. Scale bars, 50 μm. **b**–**e** The distribution of Cyclin D1, p-D1(Thr286), p-Rb(Ser780), and Ki67 determined by the staining index in the high ($n = 109$) or low ($n = 81$) DILA1-expressing groups. $n = 190$, $p$ values were determined by two-tailed Chi-square test. **f**, **g** The distribution of DILA1 (**f**) and Cyclin D1 (**g**) by the staining index in the samples of ER-positive patients with ($n = 36$) or without relapse ($n = 154$). $n = 190$, means ± s.d. are shown, $p$ values were determined by two-tailed Student's test. **h**, **i** Relapse-free survival of ER-positive breast cancer patients with low and high DILA1 (**h**) or Cyclin D1 (**i**) was analyzed by Kaplan–Meier plots. $n = 190$, $p$ values were determined by two-tailed log-rank test.

amplification or enhanced transcription of CCND1 gene by upstream signaling pathways, including ER, HER2, PI3K, and mitogen-activated protein kinase. Tremendous efforts have been made to inhibit tumor growth by targeting Cyclin D1-CDK 4/6 or its upstream pathways, with several successful drugs available in

the clinic. However, the resistance to such treatments often occurs and cancer cells that are resistant to tamoxifen[15] or CDK4/6 inhibitor[35] are still dependent on Cyclin D1 for proliferation, suggesting that alternative strategy is needed to block Cyclin D1 activity. On the other hand, Cyclin D1 is a tightly regulated

protein with the half-life of only ~24 min[31] and the dysregulated degradation also frequently leads to its overexpression[36]. Although a number of agents were shown to induce Cyclin D1 degradation in vitro[37], the mechanism was not clear and specific. Thus no effective way can be used to target Cyclin D1 degradation clinically yet. Our study showed that DILA1 specifically interacted with Cyclin D1 and regulated its degradation, indicating that DILA1 could be a specific target to control the post-translational regulation of Cyclin D1 protein.

The majority of breast cancer is ER positive and suitable for endocrine therapy, including tamoxifen. Tamoxifen can inhibit the ER signaling and the ensuing transcription of CCND1 gene, decreasing the relapse rate by 40% in ER-positive breast cancer patients. Nevertheless, the overexpression of Cyclin D1 was found to be associated with poor prognosis and tamoxifen resistance in these patients. Indeed, two tamoxifen-resistant breast cancer cell line models used in this study showed that Cyclin D1 protein was further increased in resistant cells. Interestingly, this upregulation in resistant cells was not caused by enhanced transcription by alternative upstream pathways but by suppressed degradation of Cyclin D1, indicating previously unappreciated importance of Cyclin D1 degradation in tamoxifen resistance.

We[19] and others have shown that lncRNAs can directly interact with key signaling proteins to regulate their function. It was reported that lncRNA ABHD11-AS1[38] interacted with Cyclin D1 and promoted the growth of endometrial cancer cells, but the mechanism was unclear. LncRNA was also shown to regulate the transcription of CCND1[39]. Our results indicate that DILA1 is a lncRNA that directly interacts with Cyclin D1 and regulates its degradation. More importantly, DILA1 was upregulated in tamoxifen-resistant cells and the breast cancer patients who relapsed after tamoxifen treatment, suggesting that DILA1 is a good biomarker to predict poor prognosis and tamoxifen resistance in ER-positive breast cancer patients. Knocking down DILA1 increased Cyclin D1 degradation and reversed tamoxifen resistance both in vitro and in vivo, indicating that DILA1 is a therapeutic target in regulating Cyclin D1 degradation and improve the efficacy of tamoxifen. It was suggested that combination therapies that simultaneously target CCND1 transcription of Cyclin D1 protein turnover may be helpful in treating cancer[40].

We have not explored the mechanism how DILA1 is regulated and increased in tamoxifen-resistant cells and this warrants further investigation. The strategy to efficiently target DILA1 in breast cancer remains to be optimized. Whether DILA1 plays such an important role in other types of cancer will be explored in future studies. Moreover, this study examined the role of DILA1 only in tamoxifen-resistant breast cancer cells and patients. Whether DILA1 plays a similar role in breast cancer that received the treatment of aromatase inhibitors or CDK4/6 inhibitors remains unknown and needs further study.

In summary, this study demonstrates the novel role of lncRNA DILA1 in regulating Cyclin D1 protein at the posttranslational level and leading to tamoxifen resistance of breast cancer. Different from targeting CCND1 transcription or CDK4/6, antagonizing DILA1 may be an alternative or complementary way to downregulate Cyclin D1 protein in cancer treatment.

## Methods

**Cell culture and transfection**. MCF-7 and T47D breast cancer cell lines and 293T cell lines were obtained from the ATCC (American Type Culture Collection). Both of them were authenticated using short tandem repeat multi-amplification and tested to be mycoplasma negative. MCF-7 cells and 293T cells were cultured in DMEM (Gibco, USA) medium, supplemented with 10% fetal bovine serum (FBS; Hyclone, USA). T47D cells were cultured in RPMI-1640 medium (Gibco, USA), supplemented with 10% FBS (Hyclone, USA).

Tamoxifen-resistant MCF7 and T47D cells were successfully established as previously reported[20] and were cultured in phenol red-free RPMI-1640 medium

with 10% Charcoal/Dextran Treated FBS (SH30068, Hyclone) and 3 μM 4-OH tamoxifen (H7904, Sigma).

For transfection of siRNA or ASOs, cells were plated at $2 \times 10^4$ per well in 24-well plate and transfected with specific siRNAs (100 nM; GenePharma, China) or ASOs (50 nM; RiboBio, China) mixed with RNAiMax (13778150, Invitrogen) according to the manufacturer's instructions. The sequences of siRNAs and ASOs are listed in Table S3.

The sequences of DILA1, CCND1, HA-tagged CCND1, and HA-tagged CCND1 T286A were cloned into a PCDH-Puro-vector (PCDH-Puro-DILA1, PCDH-Puro-Cyclin D1, PCDH-Puro-HA-Cyclin D1, PCDH-Puro-HA-Cyclin D1 T286A). The plasmids (10 μg) were transfected into 293T cells with pMD2.G (5 μg) and pSPAX2 (10 μg) plasmids to generate lentivirus.

For transduction of MCF-7 and T47D cells, cells were plated at $0.5 \times 10^5$ cells per well in 24-well plates and transduced with lentiviral particles with 5 μg/ml Polybrene. After 2 days, puromycin was added into the medium at a concentration of 3 μg/ml to select stably transduced cells.

**RIP and RIP-seq**. RIP was performed using the Magna RIP RNA-Binding Protein Immunoprecipitation Kit (17-700, Millipore) according to the manufacturer's instructions. Briefly, the cultured cells were lysed by RIP lysis buffer containing protease inhibitor cocktail and RNase inhibitor. Antibodies were added to the lysate and incubated with rotation at room temperature for 60 min. The magnetic beads were added to the mixture and incubated with rotation at 4 °C for 2 h. The beads were washed three times and the RNAs pulled down were extracted by TRIzol™ LS (10296028, Invitrogen) and subjected to RIP-seq or RIP-qPCR analysis.

For RIP-seq, anti-HA antibody (H9658, Sigma, dilution 1:100) was used to perform RIP reaction in MCF-7 cells with exogenous 5′-HA-tagged (MCF7-HA-D1) or untagged Cyclin D1 (MCF7-D1). The ribosomal RNAs were removed by a Low Input RiboMinus™ Eukaryote System (A15027, Invitrogen). Then the harvested RNAs were reverse transcribed into cDNA sequencing library using the KAPA Stranded RNA-Seq Library Preparation Kit (KK8400, Illumina). The quality of cDNAs was analyzed by the Agilent Bioanalyzer 2100 and then sequenced using the Illumina NextSeq 500 platform. The reads were mapped to human genome using TopHat2 and visualized on the ensemble (http://asia.ensembl.org/index.html) and UCSC browser (http://genome.ucsc.edu). The lncRNAs interacted with HA-tagged Cyclin D1 were selected according to the screening criteria: fold change of fragments per kilobase of transcript per million mapped reads was >2 and the p value was <0.05. Untagged Cyclin D1 was used as a negative control to rule out any RNAs non-specifically bound to anti-HA antibody.

**Quantitative PCR with reverse transcription**. TRIzol® Reagent (Life Technologies, USA) was used to extract total RNA from breast cancer cells. RNA was reverse-transcribed into cDNA using PrimeScript RT Master Mix (RR036A, Takara). Real-time quantitative PCR was performed using TB Green Premix Ex Taq II (RR820A, Takara) according to the manufacturer's recommendations. The reactions were carried out in the LightCycler480 system with gene-specific primers. The primers were designed on the website of Primer Bank (https://pga.mgh.harvard.edu/primerbank/) and PrimerBlast (https://www.ncbi.nlm.nih.gov/tools/primerblast/index.cgi?LINK_LOC=BlastHome). All the primer sequences used in this research are listed in Table S3.

**Cell number counting, MTT, colony formation, and EdU assays**. For cell number counting, the cultured cells were digested by 0.4% trypsin and counted using a cell counter (IC 1000, Countstar). For MTT analysis, MTT powder (3580MG250, Biofrox) was dissolved in sterile phosphate-buffered saline (PBS) at a concentration of 5 mg/ml, added into the cells cultured in 96-well plates with a 1:10 ratio. After incubation at 37 °C for 4 h, the absorbance was measured at a wavelength of 490 nm by a microplate spectrophotometer. For colony-formation assay, cells (1000/well) were planted in 6-well plates and cultured in medium with tamoxifen (3 μM) for 2 weeks, then were fixed by 4% paraformaldehyde and stained with crystal violet for colony number count. EdU proliferation was performed according to the protocol of Cell-Light EdU Apollo567 In Vitro Kit (C10310-1, RIOBIO). In brief, cells were planted in 96-well plates (3000/well) and treated with tamoxifen (3 μM) for 48 h. After the addition of EdU, the cells were cultured for 4 h, fixed by 4% paraformaldehyde, and stained by Apollo®567 and 4,6-diamidino-2-phenylindole (DAPI). Images were taken by fluorescence microscopy.

**Flow cytometry for cell cycle**. Flow cytometry was performed on a flow cytometer (Becton Dickinson) to analyze cell cycle distribution. The cells (~$10^6$) were fixed by 75% cold ethanol for >24 h and stained with 200 μl propidium iodide containing 5 μl RNAase and analyzed by flow cytometer. The gating strategy was used to exclude cell debris and aggregates (Fig. S10).

**Subcellular RNA fractionation**. Nuclear and cytoplasmic RNA was purified according to the manufacturer's recommendations of the Nuclear and Cytoplasmic RNA Purification Kit (Invitrogen, AM1921), and then the expression of DILA1 in different subcellular fractionations was analyzed by RT-qPCR.

**Western blotting**. Protein was extracted from the cells using RIPA lysis buffer containing protease and phosphatase inhibitors (78442, Thermo Scientific), separated by sodium dodecyl sulfate-polyacrylamide gel electrophoresis (SDS-PAGE) gel, transferred to polyvinylidene difluoride membranes. Membranes were incubated with primary antibodies overnight at 4 °C, followed by anti-mouse or rabbit horseradish peroxidase (HRP)-conjugated secondary antibodies (7076/7074, CST, dilution 1:1000). Afterwards, the protein–antibody complex was visualized by enhanced chemiluminescence assay (34095, Pierce). Primary antibodies against Cyclin D1 (2922, CST, dilution 1:1000), phospho-Cyclin D1 (Thr286) (3300S, CST, dilution 1:1000), GSK-3β (12456S, CST, dilution 1:1000), Rb (9309, CST, dilution 1:1000), phospho-Rb (Ser780) (ab47763, Abcam, dilution 1:1000), Lamin A/C (4777, CST, dilution 1:1000), and HRP-conjugated glyceraldehyde 3-phosphate dehydrogenase antibody (HRP-60004, Proteintech, dilution 1:5000) were used. uncropped blots are provided in supplemental information for uncropped blots and gels in supplementary information.

**Determination of DILA1 copy numbers**. Increasing numbers of in vitro transcribed DILA1 were used as standard samples for RT-qPCR, and the standard curve was generated according to DILA1 copy numbers and corresponding threshold cycle (CT) value. DILA1 in $5 \times 10^5$ cells from multiple cell lines was subjected for RT-qPCR and the DILA1 copy number per cell was determined according to their CT values compared to the standard curve and then divided by the cell number.

**Ubiquitination assay**. Cultured cells were treated with 20 μM MG132(C2211, Sigma) for 6 h and then lysed by IP lysis buffer containing protease and phosphatase inhibitors on ice for 30 min. Anti-Cyclin D1 antibody (RB-010-P, Invitrogen, dilution 1:100) or IgG was added into the lysate and incubated with rotation overnight at 4 °C. Dynabeads Protein G (10003D, Invitrogen) was added into the mixture and incubated at 4 °C for 2 h, boiled in SDS loading buffer, and used for western blotting analysis. Antibody against ubiquitin (3936, CST, dilution 1:1000) was used to detect the ubiquitination of Cyclin D1.

**5′ and 3′ rapid amplification of cDNA ends (RACE)**. 5′ and 3′ RACE was conducted using the SMARTer RACE 5′/3′ Kit (634859, Takara) according to the manufacturer's instructions. Briefly, total RNA was extracted from MCF7-Re cells, then the 5′- and 3′-RACE-Ready first-strand cDNA was synthesized with 5′- and 3′-CDS primer. The cDNAs were subjected to PCR reaction using universal primer mix with 5′ or 3′ gene-specific primers. Gene-specific primers are listed in Table S3. The amplified cDNA was purified and cloned into the p-EASY-T5-zero cloning vector (CT501-02, TransGen Biotech). The vectors were further transfected into Trans-T1 Phage Resistant chemically competent cells (CT501-03, TransGen Biotech) and amplified, then the plasmids were extracted and sequenced.

**Immunohistochemistry**. Paraffin-embedded tissue sections were dewaxed and rehydrated before antigen retrieval by boiling in 0.01 M citrate buffer (pH 6.0) for 30 min. In all, 3% hydrogen peroxide was added for 15 min to remove endogenous peroxidase. Tissues were incubated with goat serum for 30 min at room temperature and then with anti-CyclinD1 (NBP2-32840, Novus, dilution 1:100), anti-ki67 (ZM0166, ZSGB-BIO, ready to use), anti-phospho-Cyclin D1 (Thr286) (STJ90457, St John's Laboratory, dilution 1:50), and anti-phospho-Rb (Ser780) (ab47763, Abcam, dilution 1:100) antibodies at 4 °C overnight, respectively. The immunodetection was performed on the following day using DAB (GSK500710, Gene Tech) according to the manufacturer's instructions. The staining scores were determined by two independent observers, based on both the proportion and brown intensity of the indicated protein-positive cells. The proportion of positively stained tumor cells was divided into 4 grades: (0: no positive cells; 1: <10%; 2: 10–50%; and 3: >50%). The staining intensity was recorded as follows: 0 (no staining), 1 (light brown), 2 (brown), and 3 (dark brown). The SI was calculated as follows: SI = the proportion of positive cells × staining intensity. Using this method, the expression of target protein was evaluated using the SI and scored as (0, 1, 2, 3, 4, 6, or 9), with a cut-off point of <3 versus ≥3.

**RNA FISH and in situ hybridization**. Digoxin (Dig)-conjugated LNA oligonucleotide probes (5′DIG-TACAGCAATGTCAAGGCACGAT-3′DIG) were custom-made and synthesized by Exiqon (267053814, QIAGEN). Briefly, cells plated in confocal dishes ($2 \times 10^4$/dish) were fixed by 4% formaldehyde in PBS for 20 min at room temperature, digested with 0.4% trypsin for 5 min at room temperature, and permeabilized in PBS containing 0.05% Triton X-100 for 3 min on ice. For the paraffin-embedded sections, dewaxed and rehydrated tissues were digested in 10% trypsin for 40 min at room temperature. Hybridization was carried out at 54 °C overnight in hybridization solution with a probe concentration at 25 nM.

For RNA FISH, cells were incubated with anti-Dig fluorescein-conjugated antibody (13399600, Roche, dilution 1:200) overnight at 4 °C. For co-localization assay, anti-Cyclin D1 antibody (RB-010-P, Invitrogen, dilution 1:50) was added into the reaction, followed by Alexa Fluor secondary antibodies (Invitrogen, A32733, dilution 1:200) and Hoechst 33342(H3570, Invitrogen). Images were taken by confocal microscopy (LSM800, Zeiss). For ISH, tissues were incubated with anti-Dig POD-conjugated antibodies (200-032-156, Jackson, dilution 1:200) overnight

at 4 °C and stained with DAB (GSK500710, Gene Tech). The staining results were observed by two independent researchers, and the SI was calculated as described in the IHC section above.

**RNAScope**. RNAScope assay was performed to detect the single-molecule RNA using the RNAScope® Assay Kit (Advanced Cell Diagnostics, CA, USA). Fourteen paired double-Z oligonucleotide probes targeting 2–1152 nt of DILA1 were designed using the custom software (Hs-MIR99AHG-O1, NPR-0007680). The experiment was performed according to the manufacturer's instructions. In brief, cultured cells were fixed by 10% neutral formalin at room temperature for 30 min, incubated in a hydrogen peroxide solution for 15 min, and digested in protease III solution for 20 min. The cells were then hybridized with target probes at 40 °C for 2 h in a hybridization oven. After signal amplification, the cells were conjugated with TSA Plus Cy3 fluorescence at 40 °C for 30 min and blocked by the HRP blocker. Then the cells were counterstained with DAPI and observed by confocal microscopy.

**Enhanced crosslinking IP and qPCR**. eCLIP-qPCR was performed as reported before[41–43]. Briefly, cells were cultured in medium with 4-thiouridine (100 μM; T4509, Sigma) for 16 h. Then cells were washed twice by cold PBS and crosslinked with ultraviolet (365 nm,150 mJ/cm²) and then lysed with NP-40 lysis buffer (FNN0021, Invitrogen) containing protease inhibitors and 1 mM dithiothreitol (P2325, Invitrogen). RNase T1 (AM2283, Invitrogen) was added to the supernatant at a final concentration of 1 U/μl and incubated at 22 °C for 15 min. Then Cyclin D1 antibody (RB-010-P, Invitrogen, dilution 1:100) or HA-tag antibody (H9658, Sigma, dilution 1:100) was added and incubated at 4 °C with rotation overnight. Forty microliters of dynabeads Protein G (10003D, Invitrogen) was added and incubated at 4 °C for 3 h. The pellets were incubated in NP-40 lysis buffer with DNase I (18047019, Invitrogen) at a concentration of 1 U/5 μl for 15 min at 37 °C. The immunoprecipitated protein–RNA complex was eluted from the beads by heat denaturing. After SDS-PAGE and nitrocellulose membrane transfer, the 35–110 KD region (a region of 75 kDa (~220 nt of RNA) above the Cyclin D1 protein size) is excised and treated with proteinase K (25530049, Invitrogen) to isolate RNA for next qPCR analysis. Primers were designed every 200 nt with 100-nt overlapping intervals to cover the full length of DILA1, listed in Table S3.

**Northern blotting**. Northern blotting was performed using the NorthernMax™ Kit (AM1940, Invitrogen) according to the manufacturer's instructions. Briefly, total RNA was extracted using GeneJET RNA purification kits (K0731, Thermo Scientific), separated using agarose gel electrophoresis and then transferred to a positively charged nylon membrane, and crosslinked using an ultraviolet light. The Dig-labeled LNA probes complementary to DILA1 were hybridized to the membrane at 54 °C overnight, then incubated with anti-Dig AP-conjugate antibody (11093274910, Roche, dilution 1: 5000) for 30 min, added with CSPD substrate. The signal was detected by enhanced chemiluminescence assay.

**In vitro phosphorylation assay**. Glutathione S-transferase (GST) fusion protein GST-cyclin D1 was purchased from Abcam (ab85247). Biotin-labeled RNA was prepared as described in the RNA pull-down procedure. 293T cells were transfected with plasmids carrying activated V5-GSK3-β or kinase-deficient V5-GSK3-β, respectively. After 48 h of transfection, cells were washed with cold PBS and lysed by IP lysis buffer, then immunoprecipitated using anti-V5 tag antibody (R960-25, Invitrogen, dilution 1:100). V5-GSK3-β immune complexes were added into kinase buffer (9802S, CST) containing 100 mM ATP (9804S, CST) and 1 μg of recombinant GST-Cyclin D1. DILA1 or the hairpin A deletion mutant was added into the reaction mixture as indicated. The reaction mixture was incubated at 30 °C for 1 h, then boiled in SDS loading buffer at 98 °C for 10 min. Thr286-phosphorylated Cyclin D1 was detected by western blotting.

**RNA pull-down**. RNA pull-down was performed using the MAGNETIC RNA PULL-DOWN KIT (20164, Pierce) according to the manufacturer's instructions. Briefly, RNAs for in vitro experiments were transcribed using the Transcript Aid T7 High Yield RNA Synthesis Kit (0441, Thermo Scientific) according to the manufacturer's instructions and then biotinylated using the components from the PULL-DOWN KIT. For proper secondary structure formation, 1 μg of biotinylated RNA in RNA structure buffer was heated to 95 °C for 2 min, put on ice for 3 min, and then left at room temperature for 30 min. Folded biotin-labeled RNA was then added into streptavidin magnetic beads rotated at room temperature for 60 min. The beads were then added to the cell lysates or recombinant GST-Cyclin D1 protein and then incubated with rotation for 2 h at 4 °C. Then the beads were washed five times and boiled in SDS loading buffer for 10 min at 98 °C. Finally, the retrieved proteins were analyzed by western blotting.

**Animal experiment**. The animal experiments were approved by Sun Yat-Sen University laboratory animal care and use committee. Four-week-old female NOD/SCID mice were purchased from Vital River Laboratory. Mice were housed under a specific pathogen-free condition of 12 h light/12 h dark cycle in a temperature- and humidity-controlled cage and were fed ad libitum. 17β-Estrogen pellets (0.72 mg,

60-day release, innovative research of America) were implanted subcutaneously 1 week before inoculating cells. In all, $1 \times 10^7$ MCF7-Re, MCF7-Vector, or MCF7-DILA1 cells were suspended in 0.1 ml sterile PBS and orthotopically injected directly into the mammary fat pads of mice. After tumors were palpable, MCF7-Re xenograft mice were randomly divided into six groups: (1) negative control group, (2) ASO-1 group, (3) ASO-2 group, (4) tamoxifen group, (5) tamoxifen+ASO-1 group, and (6) tamoxifen+ASO-2 group. MCF7-vector xenograft mice or MCF7-DILA1 xenograft mice were each divided into two groups, with or without tamoxifen, respectively. In the tamoxifen treatment group, 1 week after inoculation, tamoxifen time-released pellet (5 g, 60-day release, innovative research of America) was implanted subcutaneously per mouse. From the next day, 5 nM ASOs per tumor were injected intratumorally every 2 days in ASO treatment group. Four weeks later, all mice were euthanized, and tumor volumes were measured. All tumors were collected for ISH and IHC staining.

**Patients and tumor specimens**. Paraffin-embedded tumor samples were obtained from 190 ER-positive female breast cancer patients (age 23–81 years, median 48 years) at the Breast Tumor Center of Sun Yat-Sen Memorial Hospital. All samples were collected with signed informed consent according to the internal review and ethics boards of Sun Yat-Sen Memorial Hospital. All patients received curative breast surgery from January 2010 to December 2016 and were confirmed as ER-positive breast cancer by postoperative pathological diagnosis. All patients received adjuvant tamoxifen treatment and regular follow-up (median follow-up time: 72 months). Detailed clinicopathological information is provided in Tables S5 and S7.

**Statistical analysis**. Statistical analyses were performed using GraphPad Prism 8.0. and IBM SPSS Statistics 25. All in vitro and animal experiment results were expressed as means ± s.d. and two-tailed Student's test were used to calculate the $p$ value. Survival curves were constructed using the K-M analysis and compared using two-tailed log-rank test. Two-tailed Spearman rank correlation analysis was done to analyze the association between DILA1 and Cyclin D1 expression. Correlation between DILA1 expression and clinical parameters was determined by two-tailed Chi-square test. Univariate and multivariate Cox proportional hazards regression model was performed by IBM SPSS Statistics 25. $p < 0.05$ was considered statistically significant.

**Reporting summary**. Further information on research design is available in the Nature Research Reporting Summary linked to this article.

## Data availability

The RIP-seq profiles are submitted to the Sequence Read Archive (SRA) database (accession codes: PRJNA611025). All other remaining data supporting the conclusions of this study are available in the article and supplementary files or from the corresponding authors upon rational request. Source data are provided with this paper.

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

## Acknowledgements

This work was funded by Natural Science Foundation of China grants (81630074, 81872141, 81702630, 81672622), Guangzhou Science and Technology plan key projects (201804020076), and Natural Science Foundation of Guangdong (2019A1515010146).

## Author contributions

Q. Liu and E.S. conceived the ideas and designed the experiments. Q.S., Yudong Li., S.L., and Y.W., performed the experiments. L.J., Z.C., M.Z., Q. Li., Ying Li, and J.W. provided the clinical samples and performed analysis. Q.S., Yujie Liu., L.J., Z.W., and H.L. analyzed the data. Q.S., Yudong Li, and Q.L. wrote the paper.

## Competing interests

The authors declare no competing interests.
