## [Peer Review File · Nature Communications]

Reviewers' Comments:

Reviewer #1:

Remarks to the Author:

This manuscript investigated the potential role of a lncRNA DILA1 in tamoxifen resistance in breast cancer via inhibition of Cyclin D1 degradation. The authors claim that upregulated Cyclin D1 protein is responsible for tamoxifen resistance in breast cancer cells and that DILA1 binds to and inhibits Cyclin D1 degradation via ubiquitin-proteasome pathway.

Unfortunately, these claims and data from other downstream experiments are questionable and a major concern. This paper is not suitable for further consideration at Nat Comm. My comments are:

1. What is the rationale for identifying lncRNAs bound to Cyclin D1? It makes no sense to identify RNA targets of Cyclin D1 because it is not a bona fide nucleic acid binding protein. If the authors think that it binds to RNA indirectly, then what is the protein that recruits DILA1 to Cyclin D1? Or what are the domains in Cyclin D1 that enable it to bind to RNA. The RIP assays they performed were using a kit that is used from cells that are not crosslinked, hence the artifactual, post-lysis interaction.
2. I am not convinced that there is an abundant lncRNA from the miR99AHG. Typically, miR HGs produce miRNAs and their primary transcripts are processed very quickly and cannot be detected by Northern blotting, unless the processing is blocked. In some cases, the miR HGs do not produce miRNAs but are expressed as stable lncRNAs. Such lncRNAs can be seen in RNA-seq data. However, when I looked at an abundant lncRNA from miR99AHG locus in ENCODE data for MCF-7 cells, there is no abundant transcript. This implies that in MCF-7 cells, miR99AHG is processed into miR-99a and there is no lncRNA from this locus, unlike what the authors showed.
3. The authors used a single 22 mer probe for the RNA-FISH experiment. This will not work even for the most abundant lncRNAs such as MALAT1 or XIST. Single molecule RNA-FISH is the norm. Also, the sequence of the probe provided by the authors does not match any sequence in the UCSC genome browser. This raises concerns on the data presented.
4. The materials and methods section is very thin. It is not clear how many experiments were done. To give an example, I could not find the final conc of ASOs and siRNAs used in this study.
5. RIP-seq was performed using anti-HA antibody to IP. The authors should use anti-cyclinD1 antibody to IP and determine the interaction of cyclinD1 and DILA1 in endogenous setting.
6. Contradictive results from Fig3d vs 3e: 3d shows that MG132 treatment don't alter the levels of cyclinD1 protein, however in Fig3e shows MG132 treatment increases cyclinD1 protein (NC lane), (I assuming the labeling on the top of this figure was not correct, TAM-/ + should be MG132-/ +), need to be addressed.
7. Fig 3h & 3i, need cyclinD1 wblot to show equal IP efficiency
8. Fig 1e: The Northern blot needs a marker to show the length of RNA.
9. Fig S2d: siRNAs work efficiently in the cytoplasm but it seems that it works in this case despite the nuclear localization of DILA1. A FISH assay upon siRNA treatment should be conducted to ensure specificity.
10. Fig 2: Since Cyclin D1 affects Rb phosphorylation, it would be a good validation of the effect of DILA1 on CyclinD1 activity.
11. Fig 2d: In order to see the effect of DILA1 on Tamoxifen resistance, the experiment should be

repeated in the presence of Tamoxifen.

Reviewer #2:

Remarks to the Author:

Shi et al. report the discovery of that the lncRNA named DILA1 is a lncRNA regulator of cyclin D1, an important oncogene in breast cancer and contributor to resistance to hormonal therapy for ER+ breast cancer. Overall, the authors provide an interesting and generally comprehensive characterization of the molecular mechanism, cellular function of DILA1, and provided some intriguing potential clinical implications for this gene in breast cancer. But, revisions and additional experimental evidence that address several concerns will be required to have sufficient confidence in these conclusions for publication at Nature Communications.

1. Details of library preparation and data analyses for RIP-seq are poorly described in the methods, limiting reproducibility. This is an important omission since this platform was used to discover DILA1 as a cyclin d1 associated RNA.
2. Formaldehyde crosslinking methods fix large protein complex on RNA and frequently identify RNA-protein interactions that are indirect. But the authors suggest that a Thr286 of cyclin D1 is required for the interaction. The authors should use CLIP-qPCR to provide validation for a direct cyclin D1 interaction with DILA1 and the requirement for Thr286 for the direct interaction. Additionally, the authors performed RIP using exogenously expressed cyclin D1. Since overexpression experiment may bind RNAs in unspecific manner, endogenous binding of cyclin D1 and DILA1 in breast cancer cells should be clarified.
3. ISH for DILA1 in fig 5b is of poor quality compared fig 5d. The background is entirely brown in both control conditions but not in the ASO conditions and nuclear localization is not clear. This is not a convincing validation that the authors have achieve DILA1 knockdown in vivo.
4. In Figure 1D and E, the authors showed that DILA1 is elevated in MCF7-Re compared with MCF7-Pa, however, the authors did not show the expression of DILA1 in T47D-Re and -Pa cells.
5. In Figure 1E, the authors have shown the DILA1 expression on northern blot. The authors should show the full size of membrane result to prove the specificity of the probes used in this experiment.
6. RNA pull down assay in Figure 4h showed that last 183nt of DILA1 are required for the binding to cyclin D1. The authors should examine whether the 183nt sequences is sufficient for cyclin D1 stabilization or nuclear localization of cyclin D1 to define the function of DILA1 more precisely.
7. In Figure 6, the authors showed that DILA1 is associated with poorer relapse-free survival in ER-positive breast cancer patients. The association of DILA1 expression in overall survival of this cohort study is also important to show the clinical significance of DILA1 in breast cancer. In addition, the authors should explain how to determine the threshold of high and low DILA1 groups.
8. In cell cycle analyses, several G2/M fractions in histograms don't fit to the actual histograms, especially in NC samples of Figure 2d, Figure S1h and S2i.
9. In Figure 3e, the authors explained that "MG132 treated MCF7-Re cells" in manuscript and figure legend, however Figure 3e showed "Tam treated MCF7-Re cells". "Tam" may be wrong.
10. The manuscript was generally clear and easy to follow but could benefit from some additional editing for grammar and syntax.

Reviewer #3:

Remarks to the Author:

In this study the authors confirm that upregulated Cyclin D1 protein can accelerate cell proliferation and result in tamoxifen resistance in breast cancer cells. They identify the LncRNA, DILA1 as a new interactor of Cyclin D1. Through a series of elegant mechanistic studies they show that DILA1 specifically binds to the Thr286 of Cyclin D1 protein to inhibit phosphorylation, leading to the decreased ubiquitination and degradation of Cyclin D1. Knocking down DILA1 was shown to decrease the expression of Cyclin D1 protein and reverse tamoxifen resistance of breast cancer cells in vitro and in vivo. Aberrant expression of DILA1 was demonstrated to associate with overexpressed Cyclin D1 protein and poor prognosis in ER+ breast cancer patients on tamoxifen treatment.

This is an interesting and well executed study which teases out the mechanisms of DILA1 inhibition of Cyclin D1 protein degradation in tamoxifen resistant breast cancer. The in vivo and clinical studies however are less well described and a number of issues should be considered.

'...inhibition of proteasome degradation by MG132 did not influence Cyclin D1 protein levels in both parental and resistant cells [.....]. These results suggest that decreased degradation is responsible for the upregulated Cyclin D1 protein in tamoxifen resistant breast cancer cells.' This statement is a little ambiguous and the rationale for this conclusion should be clarified.

Did the authors consider/investigating other post-translational modifications, e.g. SUMOylation?

In vivo studies:

Did the expression of phospho-cyclin D1 alter following treatment with DILA-ASOs or over expression of DILA? What was the growth rate of the tumours over time in each of the experimental groups? Did either DILA-ASOs or over expression of DILA have any effect on metastatic burden in vivo? The DILA insitu- hybridization images in Figure 5b and d are not convincing. The NC control images are not white balanced in a similar manner to the DILA-ASOs (Figure 5b), similarly the vector controls versus the DILA overexpression vectors (Figure 5d).

Clinical studies:

The patient population are not well described. What was the age range of the patients? What treatments did they receive, if any, in addition to tamoxifen? What was the median follow-up? How was expression of DILA and Cyclin D1 assessed? Was this by a modified H-Score? Was the scoring performed by two independent observers/clinical pathologists?

Is the association between DILA/Cyclin D1 and reduced disease free survival independent of classic clinicopathological parameters? Was COX proportional-hazards modelling performed?

Does DILA/Cyclin D1 associate with poor disease-free survival in ER negative patients? Is there an association with aromatase inhibitor treatment or is this specific to tamoxifen treatment? Does expression of DILA1 associate with sensitivity to CDK4/6 inhibitors?

Minor comments

EdU incorporation by fluorescence microscopy images are unclear in Figure 2c and g.

All figure legends are very brief and greater detail would be beneficial.

Point by point response to the reviewers' comments

Reviewer #1 (Remarks to the Author):

This manuscript investigated the potential role of a lncRNA DILA1 in tamoxifen resistance in breast cancer via inhibition of Cyclin D1 degradation. The authors claim that upregulated Cyclin D1 protein is responsible for tamoxifen resistance in breast cancer cells and that DILA1 binds to and inhibits Cyclin D1 degradation via ubiquitin-proteasome pathway.

Unfortunately, these claims and data from other downstream experiments are questionable and a major concern. This paper is not suitable for further consideration at Nat Comm. My comments are:

1. What is the rationale for identifying lncRNAs bound to Cyclin D1? It makes no sense to identify RNA targets of Cyclin D1 because it is not a bona fide nucleic acid binding protein. If the authors think that it binds to RNA indirectly, then what is the protein that recruits DILA1 to Cyclin D1? Or what are the domains in Cyclin D1 that enable it to bind to RNA. The RIP assays they performed were using a kit that is used from cells that are not crosslinked, hence the artifactual, post-lysis interaction.

We thank the reviewer for this thoughtful concern. We and others have shown that lncRNAs can directly interact with important signaling proteins and regulate their function¹⁻⁴. Some of them are not traditional DNA/RNA-binding proteins. Thus, we explored whether lncRNA can directly interact with Cyclin D1, a key player in cell cycle progression.

Classical RNA-binding proteins (RBPs) generally contain RNA-binding regions. However, recent studies have shown that many newly discovered RBPs do not show architectural similarities with classical RBPs or lack known RNA-binding domains, indicating the complexity and diversity of RNA-protein complexes^{5, 6}. Thus, some crucial RNA interacting proteins could be missed if

only based on protein sequence and structural information to predict their RNA-binding ability.

There have been multiple experimental methods to explore the interaction between RNA and protein, including RNA immunoprecipitation (RIP), enhanced crosslinking and immunoprecipitation of RNA-protein complexes (eCLIP) and RNA pull-down^{5,7}. In our study, we performed RIP-seq to screen for the RNAs binding to Cyclin D1, and then confirmed that DILA1 associated with Cyclin D1 by RIP, eCLIP and RNA pull-down. According to the reviewer's suggestion, to rule out the artificial interaction formed post-lysis, eCLIP was performed in living cells crosslinked with UV, and showed that DILA1 could be bound by Cyclin D1 and this association was protected from RNAase digestion, indicating an interaction between them (Fig. 4i). Moreover, in vitro RNA pull-down with purified recombinant Cyclin D1 protein indicated that DILA1 directly bound to Cyclin D1 protein in vitro (Fig. 1j). Importantly, RIP assay in MCF7-Re cells by anti-Cyclin D1 antibody demonstrated the interaction between Cyclin D1 and DILA1 in endogenous setting (Fig. 1e). Furthermore, RNA pull-down with truncation mutants of Cyclin D1 showed that the 257 to 296 amino acids (AA) of Cyclin D1, which consisted of a "PEST" motif, was the domain of Cyclin D1 protein that interacts with DILA1(Fig. 4k). Collectively, these results demonstrated that DILA1 directly bound to Cyclin D1, suggesting that Cyclin D1 is a newly identified non-canonical RBP.

2. I am not convinced that there is an abundant lncRNA from the miR99AHG. Typically, miR HGs produce miRNAs and their primary transcripts are processed very quickly and cannot be detected by Northern blotting, unless the processing is blocked. In some cases, the miR HGs do not produce miRs but are expressed as stable lncRNAs. Such lncRNAs can be seen in RNA-seq data. However, when I looked at an abundant lncRNA from miR99AHG locus in ENCODE data for MCF-7 cells, there is no abundant

transcript. This implies that the in MCF-7 cells, miR99AHG is processed into miR-99a and there is no lncRNA from this locus, unlike what the authors showed.

We thank the reviewer for raising an important issue. miR99AHG is a miRNA cluster host gene, which is transcribed as miRNAs and lncRNAs. As the reviewer mentioned, some lncRNAs could act as pri-miRNAs to be processed and then translocate from the nucleus to the cytoplasm to form mature miRNAs^{8,9}. It was reported that the lncRNA transcript of miR99AHG was upregulated and promoted cell proliferation in acute megakaryoblastic leukemia, indicating that lncRNA transcript of MIR99AHG could be proportionally expressed and play a key role in some situations¹⁰.

ENCODE data may not be an ideal database to search for the expression of noncoding RNAs. We searched the NONCODE database (<http://www.noncode.org/index.php>) and found that the lncRNA transcript of miR99AHG (NONCODE TRANSCRIPT ID NONHSAT190705.1) is actually expressed in the breast at the third most abundant level in Human Body Map, after ovary and thyroid (Fig. 1 for reviewers). Moreover, two GEO datasets (GSE63189 and GSE108693) also showed that lncRNA transcripts of miR99AHG were expressed in MCF7 cells.

Figure 1 for reviewers. The expression profile of miR99AHG in human body map(http://www.noncode.org/show_rna.php?id=NONHSAT190705&version=1&utd=1#)

We examined the expression of DILA1 and miR-99a-5p (AACCCGUAGAUCCGAUCUUGUG) in parental and tamoxifen resistant MCF-7 cells. Both transcripts could be detected in parental MCF-7 cells, but only DILA1 was significantly upregulated in tamoxifen-resistance MCF-7 cells (Fig. 1d, Fig. 2a for reviewers). The copy numbers of DILA1 were significantly increased in tamoxifen-resistant MCF-7 and T47D cells (385 ± 22 and 254 ± 22 copies per cell respectively) than in the parental ones (130 ± 28 and 75 ± 5 copies per cell respectively) (Fig S3d), indicating the expression of DILA1 is abundant in these cell lines. Moreover, we performed FISH, RNAscope and northern blotting to verify the existence of DILA1 (Fig. 1h, 1i and S3b, S3e).

To further exclude the role of miR-99a-5p in tamoxifen resistance, we knocked down and overexpressed miR-99a-5p by its inhibitors and mimics

respectively, and found that the decrease or increase of miR-99a-5P didn't change tamoxifen sensitivity in breast cancer (Fig. 2b-e for reviewers). Western blotting showed no significant change of the protein level of Cyclin D1, p-D1(Thr286), Rb, p-Rb(Ser780) in the cells transfected with miR-99a-5p inhibitors or mimics, suggesting that miR-99a-5p can not affect the expression of Cyclin D1 and the subsequent Rb phosphorylation(Fig. 2f for reviewers). Collectively, these results indicate that LncRNA DILA1 is the most preferentially expressed and functional transcript of MIR99AHG in the tamoxifen-resistant breast cancer cells.

Figure 2 for reviewers. miR-99a-5p did not play a key role in tamoxifen resistance.

a The expression of miR-99a-5p in MCF-7-Pa and MCF-7-Re cells assayed by qRT-PCR. **b** miR-99a-5p inhibitor or NC was transfected into MCF7-Re cells for 48 hours. RT-qPCR confirmed miR-99a-5p was efficiently knocked down or overexpressed. **c** miR-99a-5p mimic or NC was transfected into MCF7-Re cells for 48 hours. RT-qPCR confirmed miR-99a-5p was efficiently

overexpressed. **d,e** Cell counts determined the sensitivity to tamoxifen in MCF-Re cells transfected with miR-99a-5p inhibitor(**d**) or mimic(**e**). **f** Western blotting detected the protein expression of total Cyclin D1, p-D1(Thr286), total Rb and p-Rb(Ser780) in MCF-Re cells transfected with NC, miR-99a-5p inhibitor or mimic. GAPDH was used as a loading control.

We are sorry that we did not include this part of results into the revised manuscript because of the length limit. However, we would be happy to include them into the manuscript if the reviewer thinks it is necessary.

3. The authors used a single 22 mer probe for the RNA-FISH experiment. This will not work even for the most abundant lncRNAs such as MALAT1 or XIST. Single molecule RNA-FISH is the norm. Also, the sequence of the probe provided by the authors does not match any sequence in the UCSC genome browser. This raises concerns on the data presented.

We thank the reviewer for pointing out the sequence mistake. There was a mistake of a single nucleotide in the probe sequence (5'DIG-TACAGCAAT**C**TCAAGGCACGAT-3'DIG) in the previous manuscript. The actual sequence used was 5'DIG-TACAGCAAT**G**TCAAGGCACGAT-3'DIG), which is completely reverse complementary with a sequence in DILA1 (5'-ATCGTGCCTTGACATTGCTGTA-3'). We apologize for this mistake and have corrected it in the revised manuscript.

The Locked-nucleic-acid (LNA) digoxigenin probe for DILA1 used in this study, incorporating locked nucleic acid (LNA)-modified bases, was designed by Exiqon, Vedbaek, Denmark (Lot number: 267053814). LNA-modified lncRNA probes, around 20nt and locked in a C3-endo N-type sugar conformation by a 2'O and 4'C methyl bridge, have a high affinity to the complementary RNA target molecules with good stability and aqueous solubility¹¹⁻¹⁴. This type of probes, with remarkably enhanced efficiency and

specificity of in situ hybridization, has been widely used in LncRNA research reported in several high profile papers^{2, 15, 16}. Thus, the 22nt LNA probe used to detect DILA1 in this study is feasible and reasonable.

To address the reviewer's concern, we further performed RNAscope analysis in parental and tamoxifen-resistant MCF-7 and T47D cells. The results showed that DILA1 was mainly located in the nucleus and was upregulated in tamoxifen-resistant cells than the parental ones (Fig. 1i and Fig. S3e), in line with the RNA-FISH results.

4. The materials and methods section is very thin. It is not clear how many experiments were done. To give an example, I could not find the final conc of ASOs and siRNAs used in this study.

We appreciate the constructive comments from the reviewer. The number of each experiment was shown in the figure legends section of the original manuscript. We have now provided more technical details, including the final concentrations of ASOs(50nM) and siRNAs(100nM) in the materials and methods section of the revised manuscript.

5. RIP-seq was performed using anti-HA antibody to IP. The authors should use anti-cyclinD1 antibody do IP and determine the interaction of cyclinD1 and DILA1 in endogenous setting.

According to the reviewer's suggestion, we have added the RIP-qPCR data using anti-Cyclin D1 antibody and anti-IgG control in MCF-7 tamoxifen-resistant cells, indicating the interaction of CyclinD1 and DILA1 in endogenous setting (Fig. 1e).

6. Contradictive results from Fig3d vs 3e: 3d shows that MG132 treatment

don't alter the levels of cyclinD1 protein, however in Fig3e shows MG132 treatment increases cyclinD1 protein (NC lane), (I assuming the labeling on the top of this figure was not correct, TAM-/+ should be MG132-/+), need to be addressed.

We thank the reviewer for pointing out the mis-labelling. Indeed, the top label of Fig. 3e should be MG132-/+, but not the TAM-/+ in the submitted manuscript. Nevertheless, the results of Fig. 3d and 3e are not contradictory to each other because their timings are very different. In figure 3d, cells were treated with MG132 for 0 to 2hours and Cyclin D1 didn't accumulate, so that the level of Cyclin D1 in lane 5 to lane 8 remained similar. But in figure 3e, cells were treated with MG132 for 24 hours, and the level of Cyclin D1 was increased due to the inhibition of proteasome by MG132 in the NC group. We have described it more clearly in the figure legends.

7. Fig 3h &3i, need cyclinD1 wblot to show equal IP efficiency

According to the reviewer's suggestion, we have shown the immunoblot (IB) of Cyclin D1 protein immunoprecipitated with anti-Cyclin D1 antibody in this revised manuscript. In figure 3h and figure S5g, IB of Cyclin D1 was higher in tamoxifen-resistant cells than that in the parental cells, similar to the input. In figure 3i, IB-Cyclin D1 was lower in ASOs-treated cells than that in NC-treated cells, consistent with the decreased level of Cyclin D1 after DILA1 ASOs treatment. These results showed similar IP efficiency in these ubiquitination assays.

8. Fig 1e: The Northern blot needs a marker to show the length of RNA.

According to the reviewer's suggestion, we have now shown a northern blot added with a marker in Fig. S3b of the revised manuscript, indicating an

upregulated expression of DILA1 in MCF7-Re cells. The length of DILA1 was around 1000 nt, in line with the 1183 nt verified by RACE.

9. Fig S2d: siRNAs work efficiently in the cytoplasm but it seems that it works in this case despite the nuclear localization of DILA1. A FISH assay upon siRNA treatment should be conducted to ensure specificity.

During the screening of LncRNAs related to tamoxifen resistance, siRNA was used because it is a cheaper method than ASO to knock down respective LncRNAs. RT-qPCR confirmed that DILA1 was silenced to around 50% by siRNA compared with NC in Fig. S2d. According to the reviewer's suggestion, we performed RNA-FISH in Fig. S2e. It showed that the nucleus located DILA1 was efficiently reduced in DILA1-siRNAs treated cells. Agarose gel electrophoresis also verified that DILA1 was indeed knocked down by DILA1-siRNAs (Fig. S2f).

10. Fig 2: Since Cyclin D1 affects Rb phosphorylation, it would be a good validation of the effect of DILA1 on CyclinD1 activity.

Cyclin D1 regulates G1/S cell cycle progression by phosphorylating and inactivating the retinoblastoma protein (Rb). We evaluated the levels of total Rb and Ser780-phosphorylated Rb (p-Rb(Ser780)) in vitro and in vivo. In Fig. S6b and Fig. S6c, western blotting showed a higher level of p-Rb(Ser780) in tamoxifen-resistant MCF-7 and T47D cells than the parental cells, while with a similar level of total Rb protein. In Fig. S6d, without changing the level of total Rb protein, DILA1-ASOs treatment in MCF7-Re cells decreased the level of pRb(Ser780), while DILA1 overexpression in MCF7-Pa cells increased it.

In animal experiment, we performed IHC to detect the level of p-Rb(Ser780). The results showed that p-Rb(Ser780) was markedly

decreased in DILA1-ASO treatment group (Fig. 5c, 5g) or increased in DILA1 overexpressed group (Fig. S8c, 8g), compared to the NC group respectively. In clinical samples, the expression of p-Rb(Ser780) was significantly higher in relapsed breast cancer than non-relapsed ones (Fig. 6a, S9b). Moreover, the level of p-Rb(Ser780) was higher in DILA1-high group and lower in DILA1-lower group (Fig. 6d). K-M analysis also demonstrated that higher p-Rb(Ser780) was associated with lower relapse-free survival (Fig. S9d). These data further support that DILA1 regulates Cyclin D1 expression and the downstream Rb phosphorylation.

11. Fig 2d: In order to see the effect of DILA1 on Tamoxifen resistance, the experiment should be repeated in the presence of Tamoxifen.

We thank the reviewer for this insightful suggestion. Accordingly, we have repeated all the cell cycle experiments with or without tamoxifen treatment. DILA1-ASOs caused G1/S cell cycle arrest in MCF7-Re and T47D-Re cells, and tamoxifen treatment further increased the degree of G1/S cell cycle arrest (Fig. 2d and Fig. S4e). Overexpression of DILA1 in MCF7-Pa and T47D-Pa cells accelerated cell cycle progression by decreasing the percentage of G1 cells, independent of tamoxifen treatment (Fig. 2h and Fig.S4j). These results indicate that knockdown of DILA1 not only causes G1/S cell cycle arrest, but also restores the sensitivity to tamoxifen. On the other hand, overexpression of DILA1 promotes G1/S cell cycle progression and leads to tamoxifen resistance.

Reviewer #2 (Remarks to the Author):

Shi et al. report the discovery of that the lncRNA named DILA1 is a lncRNA regulator of Cyclin D1, an important oncogene in breast cancer and contributor to resistance to hormonal therapy for ER+ breast cancer. Overall, the authors

provide an interesting and generally comprehensive characterization of the molecular mechanism, cellular function of DILA1, and provided some intriguing potential clinical implications for this gene in breast cancer. But, revisions and additional experimental evidence that address several concerns will be required to have sufficient confidence in these conclusions for publication at Nature Communications.

1. Details of library preparation and data analyses for RIP-seq are poorly described in the methods, limiting reproducibility. This is an important omission since this platform was used to discover DILA1 as a Cyclin D1 associated RNA.

We thank the reviewer for pointing out the strength of this manuscript. We are sorry that the technical details are not described clearly enough in the previous manuscript. We have now provided the details of library preparation and RIP-seq data analysis in the materials and methods section of this revised manuscript (Page 18, line 524-538) as below.

Anti-HA antibody (H9658, Sigma, dilution 1:100) was used to perform RNA Immunoprecipitation reaction in MCF-7 cells with exogenous 5'-HA-tagged (MCF7-HA-D1) or untagged Cyclin D1 (MCF7-D1). The ribosomal RNAs were removed by a Low Input RiboMinus™ Eukaryote System (A15027, Invitrogen). Then the harvested RNAs were reverse transcribed into cDNA sequencing library using KAPA Stranded RNA-Seq Library Preparation Kit (KK8400, Illumina). The quality of cDNAs was analyzed by the Agilent Bioanalyzer 2100 and then sequenced using the Illumina NextSeq 500 platform, and then sequenced using the Illumina NextSeq 500 platform. The reads were mapped to human genome using TopHat2 and visualized on the ensemble (<http://asia.ensembl.org/index.html>) and UCSC browser (<http://genome.ucsc.edu>). The LncRNAs interacted with HA-tagged Cyclin D1

were selected according to the screening criteria: fold change of FPKM was higher than 2 and the p value was less than 0.05. Untagged Cyclin D1 was used as a negative control to rule out any RNAs non-specifically bind to anti-HA antibody.

2. Formaldehyde crosslinking methods fix large protein complex on RNA and frequently identify RNA-protein interactions that are indirect. But the authors suggest that a Thr286 of Cyclin D1 is required for the interaction. The authors should use CLIP-qPCR to provide validation for a direct Cyclin D1 interaction with DILA1 and the requirement for Thr286 for the direct interaction. Additionally, the authors performed RIP using exogenously expressed Cyclin D1. Since overexpression experiment may bind RNAs in unspecific manner, endogenous binding of Cyclin D1 and DILA1 in breast cancer cells should be clarified.

We thank the reviewer for this constructive suggestion. We have now performed an eCLIP assay in this revised manuscript.

According to the reviewer's suggestion, to rule out the artificial interaction formed post-lysis, eCLIP was performed in living cells crosslinked with UV. It showed that DILA1 could be bound by Cyclin D1 and this association could protect DILA1 from RNAaseT1 treatment, indicating a direct interaction between them (Fig. 4i). Moreover, in vitro RNA pull-down with purified recombinant Cyclin D1 protein also indicated that DILA1 directly bound to Cyclin D1 protein (Fig. 1j). Importantly, RIP assay in MCF7-Re cells by anti-Cyclin D1 antibody demonstrated the interaction between Cyclin D1 and DILA1 in endogenous setting (Fig. 1e).

Furthermore, RNA pull-down with truncation mutants of Cyclin D1 showed that the 257 to 296 amino acids (AA) region of Cyclin D1, which consisted of a "PEST" motif, was the domain of Cyclin D1 that interacts with DILA1(Fig. 4k).

To determine whether Thr286 of Cyclin D1 is required for its interaction with DILA1, eCLIP assay was performed using an anti-HA antibody in MCF7-Re cells transfected with HA-Cyclin D1 vector or HA-Cyclin D1(T286A) vector. DILA1 (1000-1183nt) was precipitated together with HA-tagged Cyclin D1, but not with HA-tagged Cyclin D1 mutant at threonine 286 (T286A) (Fig. 4n), indicating that threonine at 286 of Cyclin D1 is indispensable for the direct interaction with DILA1.

3. ISH for DILA1 in fig 5b is of poor quality compared fig 5d. The background is entirely brown in both control conditions but not in the ASO conditions and nuclear localization is not clear. This is not a convincing validation that the authors have achieved DILA1 knockdown in vivo.

We apologize for the poor quality of DILA1 ISH pictures in the previous manuscript. We optimized the ISH condition and provided DILA1 ISH staining pictures with better quality in Fig. 5c and Fig. S8c in the revised manuscript. DILA1 staining intensity was much lower in both DILA1-ASO groups compared to the NC group, suggesting that DILA1 was efficiently knocked down by DILA1-ASOs treatment in vivo. Consistently, DILA1 staining was higher in DILA1 overexpression group compared to the NC group with or without tamoxifen treatment, confirming that DILA1 was overexpressed successfully in vivo and tamoxifen treatment didn't affect its expression.

4. In Figure 1D and E, the authors showed that DILA1 is elevated in MCF7-Re compared with MCF7-Pa, however, the authors did not show the expression of DILA1 in T47D-Re and -Pa cells.

We are sorry for missing to provide the DILA1 expression in T47D-Re and T47D-Pa cells in the previous manuscript. The results are now added in the

revised manuscript. In Fig. S3a and S3e, RT-qPCR and RNAScope assay showed the expression of DILA1 in T47D-Re cells was significantly upregulated compared to that in T47D-Pa cells.

5. In Figure 1E, the authors have shown the DILA1 expression on northern blot. The authors should show the full size of membrane result to prove the specificity of the probes used in this experiment.

The full-size membrane of the northern blot is now in Fig. S3b of the revised manuscript, proving the specificity of the probe used.

6. RNA pull down assay in Figure 4h showed that last 183nt of DILA1 are required for the binding to Cyclin D1. The authors should examine whether the 183nt sequences is sufficient for Cyclin D1 stabilization or nuclear localization of Cyclin D1 to define the function of DILA1 more precisely.

We thank the reviewer for raising this important point. RNA pull-down and CLIP-qPCR have showed the last 183nt(1000-1183nt) of DILA1 was responsible for Cyclin D1 binding (Fig 4h-4j). The last 183nt sequence of DILA1(DILA1-S6) was overexpressed in MCF7-Pa cells by transfected with DILA1-S6 vector (Fig. S7a). Cell count indicated cells transfected with DILA1-S6 vector were more resistant to tamoxifen compared to the cells transfected with control vector (Fig. S7b).

Western blotting showed the level of Cyclin D1 protein remained high after cycloheximide (CHX) treatment for 2 hours in MCF7-Pa cells overexpressed with DILA1-S6, while Cyclin D1 protein decreased rapidly and its half-life was ~1hr in control cells (Fig. S7c). Immunoprecipitation of Cyclin D1 followed by anti-ubiquitin immunoblotting demonstrated that ubiquitinated Cyclin D1 was markedly decreased when DILA1-S6 was overexpressed (Fig.S7d). Compared

to the control cells, MCF7-Pa cells with DILA1-S6 overexpression showed upregulation of total Cyclin D1, p-Rb(Ser780) and downregulation of p-D1(Thr286) (Fig. S7e). Meanwhile, Cyclin D1 decreased in cytoplasm and increased in nucleus with this treatment (Fig. S7f). Totally, these results suggest that the last 183 nt of DILA1 (DILA1-S6) is sufficient to inhibit the phosphorylation and cytoplasmic translocation of Cyclin D1.

7. In Figure 6, the authors showed that DILA1 is associated with poorer relapse-free survival in ER-positive breast cancer patients. The association of DILA1 expression in overall survival of this cohort study is also important to show the clinical significance of DILA1 in breast cancer. In addition, the authors should explain how to determine the threshold of high and low DILA1 groups.

According to the reviewer's suggestion, we performed K-M analysis to determine the association between the expression of DILA1, Cyclin D1, pRb and overall survival of the 190 ER positive breast cancer patients' cohort. The results showed that Cyclin D1, but not DILA1 or pRb, was significantly associated with overall survival of this cohort, although DILA1 showed an instatistically significant trend ($p=0.073$) (Fig. S9e-g). This may be caused by relatively small patient number in this cohort, different treatments after relapse, etc.

ROC Curve was performed by SPSS software (IBM SPSS Statistics 25) to determine the cut-off value of high and low DILA1 expression (Fig. S9a). It showed that $SI=3$ (SI: staining index) was the most optimal score to separate DILA-high and DILA1-low group. According to the SI value of DILA1, $SI \geq 3$ was classified into DILA1-high group, while $SI < 3$ was classified into DILA1-low group.

8. In cell cycle analyses, several G2/M fractions in histograms don't fit to the actual histograms, especially in NC samples of Figure 2d, Figure S1h and S2i.

Indeed, several G2/M fractions in histograms didn't fit to the actual histograms in the previous manuscript. This is mainly caused by the difference of G2/M fractions among three independent experiments, which leads to the G2/M fractions in the represented pictures different from the ones in histograms. We have redone the cell cycle experiments with or without tamoxifen treatment as suggested by reviewer 1. The new results of cell cycle analysis in the revised manuscript were chosen to represent the overall results from three independent experiments.

9. In Figure 3e, the authors explained that "MG132 treated MCF7-Re cells" in manuscript and figure legend, however Figure 3e showed "Tam treated MCF7-Re cells". "Tam" may be wrong.

We apologize for the wrong labeling "MG132" as "TAM" on the top of Figure 3e and we had corrected it with "DMSO" and "MG132(24h)" in this revised manuscript (Fig. 3e).

10. The manuscript was generally clear and easy to follow but could benefit from some additional editing for grammar and syntax.

We thank the reviewer for the suggestion and further editing of grammar and syntax have been done.

Reviewer #3 (Remarks to the Author):

In this study the authors confirm that upregulated Cyclin D1 protein can

accelerate cell proliferation and result in tamoxifen resistance in breast cancer cells. They identify the LncRNA, DILA1 as a new interactor of Cyclin D1. Through a series of elegant mechanistic studies they show that DILA1 specifically binds to the Thr286 of Cyclin D1 protein to inhibit phosphorylation, leading to the decreased ubiquitination and degradation of Cyclin D1. Knocking down DILA1 was shown to decrease the expression of Cyclin D1 protein and reverse tamoxifen resistance of breast cancer cells in vitro and in vivo. Aberrant expression of DILA1 was demonstrated to associate with overexpressed Cyclin D1 protein and poor prognosis in ER+ breast cancer patients on tamoxifen treatment.

This is an interesting and well executed study which teases out the mechanisms of DILA1 inhibition of Cyclin D1 protein degradation in tamoxifen resistant breast cancer. The in vivo and clinical studies however are less well described and a number of issues should be considered.

1. '...inhibition of proteasome degradation by MG132 did not influence Cyclin D1 protein levels in both parental and resistant cells [.....]. These results suggest that decreased degradation is responsible for the upregulated Cyclin D1 protein in tamoxifen resistant breast cancer cells.' This statement is a little ambiguous and the rationale for this conclusion should be clarified.

Did the authors consider/investigating other post-translational modifications, e.g. SUMOylation?

We thank the reviewer for pointing out the significance and strength of this manuscript.

We are sorry that the statement regarding the MG132 and CHX treatment is a little ambiguous. It was written in the previous manuscript that "However, inhibition of proteasome degradation by MG132 did not influence Cyclin D1 protein levels in both parental and resistant cells (Fig. 3d and Fig. S6f). These results suggest that decreased degradation is responsible for the upregulated

Cyclin D1 protein in tamoxifen resistant breast cancer cells.”

To make it clearer, we changed the sentence in the revised manuscript to “However, when proteasome degradation was inhibited by MG132 for ~2hrs to see whether there is a difference of protein synthesis, Cyclin D1 levels remained steady in MCF7-Re and MCF7-Pa cells (Fig. 3d), indicating that the protein synthesis of Cyclin D1 was not different between MCF7-Re and MCF7-Pa cells. Together with similar Cyclin D1 mRNA levels in MCF7-Re and MCF7-Pa cells, these results suggest that protein degradation but not protein synthesis was responsible for the upregulated Cyclin D1 protein in tamoxifen resistant breast cancer cells.”

It has been well established that Cyclin D1 turnover is mainly governed by ubiquitination and proteasomal degradation, which is regulated by Cyclin D1 phosphorylation on threonine-286. The SCF(Fbx4/alphaB-crystallin) ubiquitin ligase recognized and bound to Thr286-phosphorylated Cyclin D1 and followed by proteasome-dependent degradation¹⁷⁻¹⁹. Since SUMOylation has not been reported to impact the stability of Cyclin D1, we appreciate the idea of other post-translational modifications such as SUMOylation may participate in this process and may pursue it further in the future study.

In vivo studies:

Did the expression of phospho-Cyclin D1 alter following treatment with DILA-ASOs or over expression of DILA? What was the growth rate of the tumours over time in each of the experimental groups? Did either DILA-ASOs or over expression of DILA have any effect on metastatic burden in vivo? The DILA insitu- hybridization images in Figure 5b and d are not convincing. The NC control images are not white balanced in a similar manner to the DILA-ASOs (Figure 5b), similarly the vector controls versus the DILA overexpression vectors (Figure 5d).

We thank the reviewer for raising important points on in vivo data. According to the suggestion, we have performed IHC staining to detect the level of Thr286-phosphorylated Cyclin D1(p-D1(Thr286)) in xenograft sections. The results showed that the expression of p-D1(Thr286) was increased in DILA1-ASOs treated samples and decreased in DILA1 overexpressed samples (Fig. 5c,5f and S8c,8f).

We have added the growth curve of xenografts in Fig. 5a and Fig. S8a. The results showed that DILA1-ASOs inhibited the tumor growth and restored the sensitivity to tamoxifen in tamoxifen resistance tumors (Fig. 5a), while tumors with overexpressed DILA1 grew larger than vector control and were resistant to tamoxifen treatment (Fig. S8a).

We sincerely apologize that we did not look at the metastatic burden in vivo because this study mainly focused on tamoxifen resistance. It is known that ER+ MCF7 and T47D xenografts are weakly metastatic. It was reported that the bone metastasis formation by T47D cells (5×10^5 cells injected into the left cardiac ventricle of mouse) took longer than 70 days²⁰. MCF7 and T47D cells inoculated into the mammary fat pads of NOD/SCID mice may take even longer time to cause metastasis. However, cancer cells with overexpressed DILA1 grow faster and may form metastasis in a shorter time. We may explore the impact of DILA1 on the metastasis process in future studies.

We are sorry for the poor quality of DILA1 ISH pictures in the previous manuscript. We optimized the ISH condition and provided DILA1 ISH staining pictures with better quality in Fig. 5c and Fig. S8c in the revised manuscript. DILA1 staining intensity was much lower in both DILA1-ASO groups compared to the NC group, indicating that DILA1 was efficiently knocked down by DILA1-ASOs treatment in vivo. Consistently, DILA1 staining was higher in the DILA1 overexpression group compared to the NC group with or without tamoxifen treatment, confirming that DILA1 was overexpressed successfully in

vivo and tamoxifen treatment didn't affect its expression.

Clinical studies:

The patient population are not well described. What was the age range of the patients? What treatments did they receive, if any, in addition to tamoxifen? What was the median follow-up?

We apologize for missing this data and have now described the characteristics of patients in the material and method section of this revised manuscript.

In this cohort, 190 ER positive women breast cancer patients (age 23-81 years, median 48 years) were enrolled. All patients received breast surgical treatment from January 2010 to December 2016 and were confirmed as ER positive breast cancer by postoperative pathological diagnosis. All patients received adjuvant tamoxifen treatment and were regularly followed-up, with a median follow-up at 72 months. More clinicopathological information is provided in Table S5 and Table S7.

How was expression of DILA and Cyclin D1 assessed? Was this by a modified H-Score? Was the scoring performed by two independent observers/clinical pathologists?

As described in the material and method section of the revised manuscript, the staining scores were determined by two independent observers, based on both the proportion and brown intensity of indicated protein-positive cells. The proportion of positively stained tumor cells was divided into 4 grades (0: no positive cells; 1: <10%; 2: 10–50%; and 3:>50%). The staining intensity was recorded as follows: 0 (no staining), 1 (light brown), 2 (brown), and 3 (dark brown). The staining index (SI) was calculated as follows: SI =the proportion of positive cells × staining intensity. Using this method, the expression of target protein was evaluated using the SI and scored as (0, 1, 2, 3, 4, 6, or 9).

Is the association between DILA/Cyclin D1 and reduced disease-free survival independent of classic clinicopathological parameters? Was COX proportional-hazards modelling performed?

We thank the reviewer for the insightful suggestion. Univariate and multivariate Cox proportional hazards regression analysis have been done to analyze whether DILA1, Cyclin D1 and the clinical parameters were independent factors to predict relapse-free survival in 190 ER positive women breast cancer patients. It indicated that only DILA1 expression was an independent prognosis predictor in this ER positive breast cancer cohort (Table S7).

Does DILA/Cyclin D1 associate with poor disease-free survival in ER negative patients? Is there an association with aromatase inhibitor treatment or is this specific to tamoxifen treatment? Does expression of DILA1 associate with sensitivity to CDK4/6 inhibitors?

We appreciate these clinically important suggestions. We explored the expression of Cyclin D1 in the Curtis and TCGA dataset and found that Cyclin D1 mRNA in ER+ breast cancer patients were significantly higher than in ER- patients (Fig. 3a,b for reviewers). Moreover, Kaplan–Meier survival curve analysis using Curtis and K-M plotter dataset showed that higher Cyclin D1 expression was associated with shorter overall survival and relapse-free survival only in ER positive breast cancer patients (Fig.3c,d for reviewers), but not in ER negative ones (Fig. 3e,f for reviewers), indicating that Cyclin D1 expression is mainly important for the prognosis of ER positive breast cancer patients. Thus, this study was focused on the role of Cyclin D1 and DILA1 only in ER positive breast cancer cell lines and patients.

We examined the expression of DILA1 in long-term estrogen-deprived (LTED, similar to aromatase inhibitor-resistance in clinic because aromatase

inhibitor worked indirectly on breast cancer by decreasing estrogen) and parental MCF-7 cells by RT-qPCR. The results showed that DILA1 expression was similar in MCF7-Pa and MCF7-LTED cells, indicated DILA1 may not play a key role in the sensitivity to aromatase inhibitor (Fig.4a for reviewers). Because more than half of breast cancer patients in China are pre-menopausal or peri-menopausal and aromatase inhibitors were not used as often as tamoxifen, we could not find sufficient patient samples to examine the association between DILA1 expression and the sensitivity to aromatase inhibitor yet.

We examined the sensitivity to CDK4/6 inhibitors (Palbociclib) in MCF-7 or T47D cells transfected with DILA1 vector or control vector. It showed that DILA1 overexpression caused the resistance to Palbociclib (Fig. 5a,b for reviewers). It had been reported that knockdown of Cyclin D1 enhanced the sensitivity to CDK4/6 inhibitors, indicated that Cyclin D1 was necessary for CDK4/6 inhibitors resistance²¹. Thus, the resistance to CDK4/6 inhibitors mediated by DILA1 overexpression in ER positive cells could be explained by the stabilization and upregulation of Cyclin D1. Since palbociclib was only introduced into China in 2018, we could not obtain the clinical samples to analyze the association between DILA1 expression and the sensitivity to palbociclib. Nevertheless, this interesting point may be addressed in future studies.

Although these questions are clinically important, we think these results may be not necessary for this manuscript. We have another manuscript under preparation comparing the expression of different genes including Cyclin D1 in ER+ vs ER- breast cancer. The results in LTED cells and palbociclib are also not comprehensive enough. Since this manuscript is mainly about DILA1 in regulating Cyclin D1 and tamoxifen resistance, we suggest not include these results in this manuscript. However, if the reviewer thinks it is necessary, we would be happy to include them into the manuscript.

Figure 3 for reviewers. Highly expression of CyclinD1 predicts poor prognosis in ER+ rather than ER- breast cancer

a, b Comparison of CCND1 mRNA expression between ER positive and ER negative breast cancer tissues from Curtis(**a**) and TCGA(**b**) dataset published on oncomine platform. P values (***) $p < 0.01$ were determined by Student's t test.

c, Kaplan-Meier plots of overall survival for ER positive breast cancer patients from Curtis. **d**, Kaplan-Meier plots of relapse-free survival for ER positive

breast cancer patients from K-M plotter dataset. **e**, Kaplan-Meier plots of overall survival for ER negative breast cancer patients from Curtis. **f**, Kaplan-Meier plots of relapse-free survival for ER negative breast cancer patients from K-M plotter dataset. For **c-d**, Patient groups were separated based on the high or low expression of CCND1. P values were determined by the log-rank test.

Figure 4 for reviewers. **a** RT-qPCR showed the DILA1 level in parental and aromatase inhibitor-resistance MCF-7 cells. MCF7-Pa: parental MCF-7 cells; MCF7-LTED: aromatase inhibitor-resistance MCF-7 cells.

Figure 5 for reviewers. DILA1 overexpression did not affect the sensitivity to CDK4/6 inhibitors **a,b** MCF7-Pa or T47D-Pa cells were transfected with DILA1 vector or control vector for 48 hours, cell count determined the sensitivity to Palbociclib (pal).

Minor comments

EdU incorporation by fluorescence microscopy images are unclear in Figure 2c and g.

We have provided the EdU incorporation by fluorescence microscopy images with better quality in this revised manuscript.

All figure legends are very brief and greater detail would be beneficial.

We tried to limit the length of figure legends in the previous manuscript and now the figure legends in the revised manuscript were supplemented with more details.

References

1. Liu, B. *et al.* A cytoplasmic NF-kappaB interacting long noncoding RNA blocks IkappaB phosphorylation and suppresses breast cancer metastasis. *Cancer cell* **27**, 370-381 (2015).
2. Chen, F. *et al.* Extracellular vesicle-packaged HIF-1alpha-stabilizing lncRNA from tumour-associated macrophages regulates aerobic glycolysis of breast cancer cells. *Nature cell biology* **21**, 498-510 (2019).
3. Luo, M.L. *et al.* The Role of APAL/ST8SIA6-AS1 lncRNA in PLK1 Activation and Mitotic Catastrophe of Tumor Cells. *Journal of the National Cancer Institute* **112**, 356-368 (2020).
4. Guan, H. *et al.* Long noncoding RNA LINC00673-v4 promotes aggressiveness of lung adenocarcinoma via activating WNT/beta-catenin signaling. *Proceedings of the National Academy of Sciences of the United States of America* **116**, 14019-14028 (2019).
5. Ramanathan, M., Porter, D.F. & Khavari, P.A. Methods to study

- RNA-protein interactions. *Nature methods* **16**, 225-234 (2019).
6. Castello, A. *et al.* Comprehensive Identification of RNA-Binding Domains in Human Cells. *Molecular cell* **63**, 696-710 (2016).
 7. Ferre, F., Colantoni, A. & Helmer-Citterich, M. Revealing protein-lncRNA interaction. *Briefings in bioinformatics* **17**, 106-116 (2016).
 8. Beermann, J., Piccoli, M.T., Viereck, J. & Thum, T. Non-coding RNAs in Development and Disease: Background, Mechanisms, and Therapeutic Approaches. *Physiological reviews* **96**, 1297-1325 (2016).
 9. Saini, H.K., Griffiths-Jones, S. & Enright, A.J. Genomic analysis of human microRNA transcripts. *Proceedings of the National Academy of Sciences of the United States of America* **104**, 17719-17724 (2007).
 10. Emmrich, S. *et al.* LincRNAs MONC and MIR100HG act as oncogenes in acute megakaryoblastic leukemia. *Molecular cancer* **13**, 171 (2014).
 11. Lignon, G., Hotton, D., Berdal, A. & Bolanos, A. In Situ Hybridization in Mineralized Tissues: The Added Value of LNA Probes for RNA Detection. *Methods in molecular biology* **1922**, 181-190 (2019).
 12. Soares, R.J. *et al.* Evaluation of fluorescence in situ hybridization techniques to study long non-coding RNA expression in cultured cells. *Nucleic acids research* **46**, e4 (2018).
 13. Fontenete, S. *et al.* Application of locked nucleic acid-based probes in fluorescence in situ hybridization. *Applied microbiology and biotechnology* **100**, 5897-5906 (2016).
 14. Kubota, K., Ohashi, A., Imachi, H. & Harada, H. Improved in situ hybridization efficiency with locked-nucleic-acid-incorporated DNA probes. *Applied and environmental microbiology* **72**, 5311-5317 (2006).
 15. Lin, A. *et al.* The LINK-A lncRNA activates normoxic HIF1 α signalling in triple-negative breast cancer. *Nature cell biology* **18**, 213-224 (2016).
 16. Wang, P. *et al.* Long noncoding RNA lnc-TSI inhibits renal fibrogenesis

- by negatively regulating the TGF-beta/Smad3 pathway. *Science translational medicine* **10** (2018).
17. Alt, J.R., Cleveland, J.L., Hannink, M. & Diehl, J.A. Phosphorylation-dependent regulation of cyclin D1 nuclear export and cyclin D1-dependent cellular transformation. *Genes & development* **14**, 3102-3114 (2000).
 18. Barbash, O., Egan, E., Pontano, L.L., Kosak, J. & Diehl, J.A. Lysine 269 is essential for cyclin D1 ubiquitylation by the SCF(Fbx4/alphaB-crystallin) ligase and subsequent proteasome-dependent degradation. *Oncogene* **28**, 4317-4325 (2009).
 19. Barbash, O. *et al.* Mutations in Fbx4 inhibit dimerization of the SCF(Fbx4) ligase and contribute to cyclin D1 overexpression in human cancer. *Cancer Cell* **14**, 68-78 (2008).
 20. Gawrzak, S. *et al.* MSK1 regulates luminal cell differentiation and metastatic dormancy in ER(+) breast cancer. *Nature cell biology* **20**, 211-221 (2018).
 21. Kong, T. *et al.* eIF4A Inhibitors Suppress Cell-Cycle Feedback Response and Acquired Resistance to CDK4/6 Inhibition in Cancer. *Molecular cancer therapeutics* **18**, 2158-2170 (2019).

Reviewers' Comments:

Reviewer #1:

Remarks to the Author:

While some comments were addressed, I am not convinced by the response that "We and others have shown that lncRNAs can directly interact with important signaling proteins and regulate their function". Some lncRNAs can interact indirectly with signaling proteins via adaptor proteins. This has been discussed in Section 3.1 in a recent review article by John Rinn and Howard Chang that was published in *Annu. Rev. Biochem* in 2020.

This raises concerns on the scientific premise and mechanism proposed in this study.

Reviewer #2:

Remarks to the Author:

The authors have adequately addressed my comments and I support accepting this manuscript for publication.

Reviewer #3:

Remarks to the Author:

The authors have addressed the concerns raised from the original manuscript with the exception of determining if the association with DILA1 and disease progression is specific to tamoxifen treated patients or is it also pertinent to AI treated patients. Nor have they assessed the impact of DILA1 on CDK4/6 inhibitor treatment. As the authors state that the results they have generated assessing the significance of DILA1 in LTED cells and with Palbociclib are not comprehensive they should not be included in the manuscript. The limitation that this study is restricted to tamoxifen treated models and patient data and has not examined AI treated models should be clearly stated in the discussion.

Point by point response to reviewers' comments

Reviewer #1 (Remarks to the Author):

While some comments were addressed, I am not convinced by the response that "We and others have shown that lncRNAs can directly interact with important signaling proteins and regulate their function". Some lncRNAs can interact indirectly with signaling proteins via adaptor proteins. This has been discussed in Section 3.1 in a recent review article by John Rinn and Howard Chang that was published in *Annu. Rev. Biochem* in 2020.

This raises concerns on the scientific premise and mechanism proposed in this study.

In the section 3.1 of the above-mentioned review, the author discussed that lncRNA Xist interacted with ~81 proteins, 10 of which are direct RNA–protein interactions, and the remainder are adaptor proteins-mediated RNA–protein interactions. While Xist is a traditional example of the modular organization of lncRNAs, many recent studies have shown that lncRNAs (AGPG¹, HISLA², LINK-A³) can directly interact with non-canonical RBPs (PFKFB3¹, HIF-1 α ², PIP3³). These newly discovered RBPs do not show architectural similarities with classical RBPs or lack known RNA-binding domains, indicating the complexity and diversity of RNA-protein complexes.

Using multiple methods including RIP, CLIP and RNA pull-down, we demonstrated that DILA1 interacts with Cyclin D1 in both endogenous setting and *in vitro* setting. Thus, we believe that Cyclin D1 is a newly identified noncanonical RNA-binding protein that directly interacts with DILA1.

Reviewer #2 (Remarks to the Author):

The authors have adequately addressed my comments and I support accepting this manuscript for publication.

We thank the reviewer for the positive comment.

Reviewer #3 (Remarks to the Author):

The authors have addressed the concerns raised from the original manuscript with the exception of determining if the association with DILA1 and disease progression is specific to tamoxifen treated patients or is it also pertinent to AI treated patients. Nor have they assessed the impact of DILA1 on CDK4/6 inhibitor treatment. As the authors state that the results they have generated assessing the significance of DILA1 in LTED cells and with Palbociclib are not comprehensive they should not be included in the manuscript. The limitation that this study is restricted to tamoxifen treated models and patient data and has not examined AI treated models should be clearly stated in the discussion.

We thank the reviewer for allowing us to not include the preliminary results in LTED cells and Palbociclib into the manuscript. As suggested, we have now stated this limitation in the discussion section of this revised manuscript (Page 17, Line 481-485) as below:

Moreover, this study examined the role of DILA1 only in tamoxifen-resistant breast cancer cells and patients. Whether DILA1 plays a similar role in breast cancer that received the treatment of aromatase inhibitors or CDK4/6 inhibitors remains unknown and needs further study.

1. Liu, J. *et al.* Long noncoding RNA AGPG regulates PFKFB3-mediated

tumor glycolytic reprogramming. *Nature communications* **11**, 1507 (2020).

2. Chen, F. *et al.* Extracellular vesicle-packaged HIF-1alpha-stabilizing lncRNA from tumour-associated macrophages regulates aerobic glycolysis of breast cancer cells. *Nature cell biology* **21**, 498-510 (2019).
3. Lin, A. *et al.* The LINK-A lncRNA interacts with PtdIns(3,4,5)P3 to hyperactivate AKT and confer resistance to AKT inhibitors. *Nature cell biology* **19**, 238-251 (2017).